# Clinical impact of statin intensity according to age in patients with acute myocardial infarction

**Kyusup Lee**[1,2], **Myunhee Lee**[1,2], **Dae-Won Kim**[1,2], **Jinseob Kim**[3], **Sungmin Lim**[1,4], **Eun Ho Choo**[1,5], **Chan Joon Kim**[1,4], **Chul Soo Park**[1,6], **Hee Yeol Kim**[1,7], **Ki-Dong Yoo**[1,8], **Doo Soo Jeon**[1,9], **Kiyuk Chang**[1,5], **Ho Joong Youn**[1,5], **Wook-Sung Chung**[1,5], **Min Chul Kim**[10], **Myung Ho Jeong**[10], **Youngkeun Ahn**[10], **Jongbum Kwon**[11], **Mahn-Won Park**[1,2]*

**1** Cardiovascular Research Institute for Intractable Disease, College of Medicine, The Catholic University of Korea, Seoul, Republic of Korea, **2** Department of Cardiology, Daejeon St. Mary's Hospital, College of Medicine, The Catholic University of Korea, Daejeon, Republic of Korea, **3** Department of Epidemiology, School of Public Health, Seoul National University, Seoul, Republic of Korea, **4** Division of Cardiology, Department of Internal Medicine, Uijeongbu St Mary's Hospital, The Catholic University of Korea, Uijeongbu, Republic of Korea, **5** Division of Cardiology, Department of Internal Medicine, Seoul St. Mary's Hospital, The Catholic University of Korea, Seoul, Republic of Korea, **6** Division of Cardiology, Department of Internal Medicine, Yeouido St. Mary's Hospital, The Catholic University of Korea, Seoul, Republic of Korea, **7** Division of Cardiology, Department of Internal Medicine, Bucheon St. Mary's Hospital, The Catholic University of Korea, Bucheon, Republic of Korea, **8** Division of Cardiology, Department of Internal Medicine, St. Vincent's hospital, The Catholic University of Korea, Suwon, Republic of Korea, **9** Division of Cardiology, Department of Internal Medicine, Incheon St. Mary's Hospital, The Catholic University of Korea, Incheon, Republic of Korea, **10** Cardiovascular Center, Chonnam National University Hospital, Chonnam National University, Gwangju, Republic of Korea, **11** Department of Thoracic and Cardiovascular Surgery, Daejeon St. Mary's Hospital, the Catholic University of Korea, Daejeon, Republic of Korea

* pmw6193@catholic.ac.kr

**Data Availability Statement:** The minimal anonymized data set necessary to replicate this study findings is uploaded as Supporting Information.

## Abstract

### Background

The available data are not sufficient to understand the clinical impact of statin intensity in elderly patients who undergo percutaneous coronary intervention (PCI) due to acute myocardial infarction (AMI).

### Methods

Using the COREA-AMI registry, we sought to compare the clinical impact of high- versus low-to-moderate-intensity statin in younger (<75 years old) and elderly (≥75 years old) patients. Of 10,719 patients, we included 8,096 patients treated with drug-eluting stents. All patients were classified into high-intensity versus low-to-moderate-intensity statin group according to statin type and dose at discharge. The primary end point was target-vessel failure (TVF), a composite of cardiovascular death, target-vessel MI, or target-lesion revascularization (TLR) from 1 month to 12 months after index PCI.

### Results

In younger patients, high-intensity statin showed the better clinical outcomes than low-to-moderate-intensity statin (TVF: 79 [5.4%] vs. 329 [6.8%], adjusted hazard ratio [aHR] 0.76;

**Funding:** The authors received no specific funding for this work.

**Competing interests:** The authors have declared that no competing interests exist.

95% confidence interval [CI] 0.59–0.99; P = 0.038). However, in elderly patients, the incidence rates of the adverse clinical outcomes were similar between two statin-intensity groups (TVF: 38 [11.4%] vs. 131 [10.6%], aHR 1.1; 95% CI 0.76–1.59; P = 0.63).

## Conclusions

In this AMI cohort underwent PCI, high-intensity statin showed the better 1-year clinical outcomes than low-to-moderate-intensity statin in younger patients. Meanwhile, the incidence rates of adverse clinical events between high- and low-to-moderate-intensity statin were not statistically different in elderly patients. Further randomized study with large elderly population is warranted.

## Background

For patients who experienced acute myocardial infarction (AMI), a pharmacologic therapy to prevent a recurrent cardiovascular events is a cornerstone of treatment [1]. Statin therapy has been shown to reduce major vascular events and vascular mortality in those individuals [2,3]. Thus, current guidelines recommend that high-intensity statin should be considered based on lowering low-density lipoprotein cholesterol (LDL-C) for secondary prevention in patients with atherosclerotic cardiovascular disease (ASCVD) [4–7].

Because of globally increasing number of adults aged 75 years or older and inherent their high cardiovascular risk and frailty [8–10], optimal treatment strategy is more required for elderly patients who underwent percutaneous coronary intervention (PCI) due to AMI. Recently, the European Society of Cardiology (ESC) and European Atherosclerosis Society (EAS) guideline recommends that, in older patients (>65 years old), maximal tolerated statin therapy be considered in the same way as for younger patients [11]. The American College of Cardiology (ACC)/American Heart Association (AHA) recommends that it is reasonable to continue high-intensity statin therapy after evaluation of the potential benefits versus adverse effect of statin therapy in patients older than 75 years of age with atherosclerotic cardiovascular disease (ASCVD) who are tolerating high-intensity statin therapy [12]. The National Lipid Association (NLA) recommends moderate- or high-intensity statin therapy for patients who are 75–80 years old and moderate-intensity statin therapy for patients who are older than 80 years of age [13].

However, prescription rates of statin have been shown to be declined with increasing age, and are lower, especially in elderly patients (>75 years old) [14]. Furthermore, there has been a lack of data regarding the efficacy and safety of high-intensity statin in the elderly patients (≥75 years old), mainly because this patient group is often underrepresented in randomised controlled trials comparing statin intensity. For this reason, the level of evidence of using a high-intensity statin for elderly patients is lower compared to that for younger patients in current guidelines [11,12], of which elderly patients were divided the age group by 75-years old. Moreover, there are no studies comparing the efficacy of statin intensity according to age in AMI patients treated by PCI using drug-eluting stents (DESs).

Therefore, we sought to (1) investigate the prescription rate of high-intensity statin and (2) compare the association between statin-intensity (high-intensity versus low-to-moderate-intensity statin) and clinical outcomes according to age (<75 or ≥75 years old) using large prospective AMI registry.

## Methods

### Study design and population

COREA-AMI (CardiOvascular Risk and idEntificAtion of potential high-risk population in Acute Myocardial Infarction) registry is a prospective, multicenter registry including AMI patients underwent PCI between January 2004 and August 2014. It was designed to evaluate the long-term clinical outcomes in all consecutive patients with AMI at nine major cardiac centers in Korea. All hospitals were high-volume centers undergoing PCI and are located throughout the country. Clinical, angiographic and follow-up data of all AMI patients were consecutively registered in the electronic, web-based case report system. Of 10,719, we included 8,096 patients who underwent PCI using DES and had available information of statin therapy at discharge. However, patients who treated with bare-metal stent, plain old balloon angioplasty, or thrombectomy alone, and patients who experienced in-hospital death were excluded (Fig 1). All patients were classified into two groups by statin type and dose at discharge; high-intensity and low-to-moderate intensity statin group, based on American College of Cardiology/American Heart Association Classification of Intensity [15]. High-intensity statin included 40-80mg of atorvastatin or 20-40mg of rosuvastatin. Others were classified into low-to-moderate-intensity statin. We compared the efficacy on clinical outcomes according to statin-intensity in younger (<75 years old) and elderly (≥75 years old) patients, respectively.

The choice of treatment strategy including statin intensity was at the discretion of the attending cardiologists with careful consideration of clinical risk factors, anatomical complexity, and procedural characteristics. All procedure was guided by standard techniques and

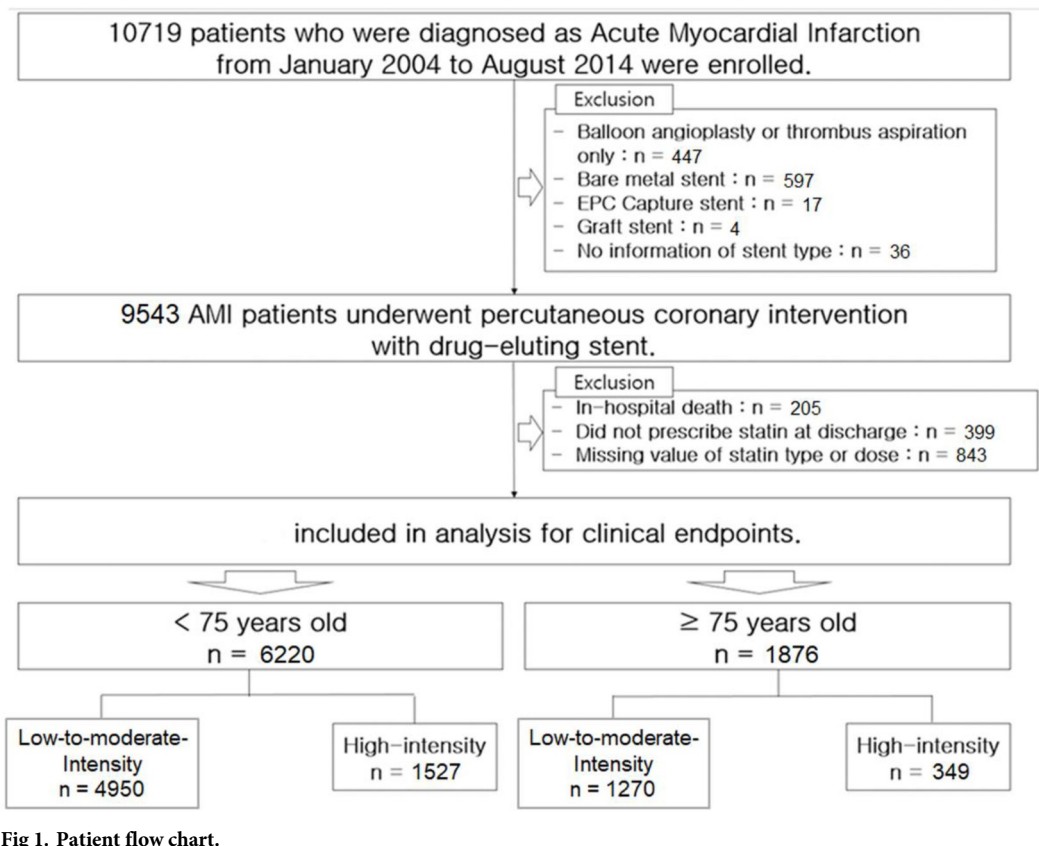

**Fig 1. Patient flow chart.**

management. Antiplatelet therapy and peri-procedural anticoagulation were performed in accordance with the accepted guidelines. During follow-up, patient management including medical treatment was performed in accordance with accepted guidelines and established standards of care. This study was approved by the local ethics committee of each hospital, and all patients provided written informed consent for the use of their clinical data for the registry study.

## Study outcomes and follow-up

The primary end point of the study was target-vessel failure (TVF), defined as a composite of cardiovascular (CV) death, target-vessel MI (TV-MI), or target lesion revascularization (TLR). The secondary end points included the individual components of primary end point, and any death. Cause of death was considered to be cardiovascular-related, unless an unequivocal non-cardiovascular cause could be established. AMI is based on the third universal definition of myocardial infarction [16]. TV-MI was defined as newly developed "AMI" due to treated vessel after discharge. TLR was defined as any percutaneous or surgical revascularization of the treated lesion. Complex PCI was defined as one of the followings: 1) bifurcation lesion; 2) chronic total occlusion (CTO) lesion; 3) PCI for left main lesion; 4) multivessel disease; 5) restenosis lesion; 6) long lesion (total length of stents ≥60 mm); and 7) number of implanted stents ≥3. All clinical events were confirmed by source documentation collected at each hospital and centrally adjudicated by an independent group of clinicians unaware of the revascularization type.

Clinical follow-up was performed at 1 month, 6 month, 12 month, and annually after index PCI. We performed a 1-month landmark analysis separated by acute (<1 month after index PCI) and maintenance phase (from 1 month to 12 months after index PCI) to estimate accurate efficacy of statin therapy. In acute phase of AMI, the ischemic complication (ie, cardiac death, MI or stent thrombosis etc...) frequently occurs in spite of optimal medical treatment including statin, because the status of patients suffering from AMI is still unstable and several factors may affect the early period of clinical outcomes. Thus, we decided to perform 1-month landmark analysis to avoid this issue of bias and figure out the effect of statin-intensity on stabilized patients who experienced AMI.

## Statistical analyses

Continuous variables were presented as mean ± standard deviation and compared using Student *t* test or Mann-Whitney *U* test. Categorical variables were presented as counts (percentages) and compared using the chi-square or Fisher exact test, as appropriate. Event rates were estimated on Kaplan–Meier estimates in time-to-first-event analyses and were compared using the log-rank test. The Cochran-Armitage trend test was used to determine difference in trends in incidence changes of event rates between acute and maintenance phase.

Clinical outcomes were evaluated between high- versus low-to-moderate-intensity statin and stratified by the age of 75. Crude and adjusted risks for clinical outcomes were compared by univariate and multivariable Cox proportional hazards regression analyses. Multiple regression analyses using Cox proportional hazard models were performed with low-to-moderate-intensity group as the reference category and with high-intensity group as the indicator variable. Variables with P values of ≤0.1 and clinically relevant covariates irrespective of their statistical relevance in univariate analyses were candidates for inclusion in multivariate Cox proportional hazards models.

Inverse probability of treatment weighting (IPTW) based on the propensity score (probability of receiving high-intensity statin, PS) was used as the primary tool to adjust for differences

in the baseline characteristics between the high-intensity and low-to-moderate intenstiy groups. Once each patient's PS was estimated, weights were calculated using the method described in the previous literature [17]. We examined the similarity of the baseline characteristics between the treatment groups before and after IPTW [18]. After confirming the comparability of the two groups in the data with IPTW, we ran the Cox proportional hazard model and made statistical inference using robust standard errors (Huber sandwich estimator) [19].

For all crude, multivariable-adjusted, and IPTW analyses, treatment effects were evaluated in overall patients and in each group of younger and elderly patients. To test the statistical significance of the difference in treatment effect of statin-intensity between younger and elderly patients, the interaction term between age (<75 or ≥75 years old) and statin intensity was included in the multivariate that were built basis of the data from all the patients. As the sensitivity analyses, we conducted propensity matched analysis to adjust for potential confounders with a logistic regression between two statin-intensity groups. The PS was calculated from the variables including age, gender, presence of diabetes mellitus, prior MI, prior coronary bypass grafting surgery (CABG), prior cerebrovascular attack, left main disease, complex PCI, and baseline value of LDL-cholesterol. Using the PS, patients were selected by 1:1 matching without replacement using the nearest-neighbor method. A caliper width of 0.2 standardized differences (SD) was used for matching. All reported p values were two-sided, and p values <0.05 were considered statistically significant. R 4.0.2 (R Foundation for Statistical Computing, Vienna, Austria.) was used for all statistical analyses.

## Results

### Baseline patients, lesions, and procedural characteristics

Baseline characteristics of overall patients were presented in Table 1. Elderly patients (≥75 years old) accounted for 20.0% (n = 1,619) of study population. Among overall patients, 6220 (76.8%) patients were in low-to-moderate-intensity statin group, and 1876 (23.2%) patients were in high-intensity statin group. This pattern in proportion of statin-intensity was observed from younger and elderly patients.

In younger patients (<75 years old), mean age was lower (57.8 ± 10.0 vs. 58.7 ± 10.2, p<0.001) and the proportion of male gender were higher (81.8% vs. 78.6%, p = 0.007) in high-intensity statin group than low-to-moderate-intensity statin group. The high-intensity statin group had a significantly greater prevalence of hyperlipidemia and lesser prevalence of diabetes mellitus, prior MI, prior CABG, and atrial fibrillation compared with low-to-moderate intensity statin group. As the discharge medication, patients in the high-intensity statin group were more likely to receive potent P2Y12 inhibitors (ticagrelor or prasugrel) than low-to-moderate-intensity statin group. In elderly patients, baseline demographic characteristics did not differ between two statin-intensity groups. In overall patients, mean stent diameter, total stent number and length was similar between the two statin-intensity groups, that was consistent regardless of age. Involvement of proximal left anterior descending artery was shown frequently in high-intensity statin group among younger patients (Table 1). As shown in Table 2 and Table 2 in S1 Appendix, majority of those differences of covariates were balanced after PS matching and IPTW.

### Incidence rates of adverse clinical outcomes among overall patients

In overall patients, the event rates of clinical outcomes during 1 year and adjusted hazard ratio (aHR) for risk of high-intensity statin compared with low-to-moderate intensity statin are shown in Table 1 in S1 Appendix. Adverse clinical outcomes were similar between two statin-intensity groups except for TLR, which was significantly lower in the high-intensity statin

**Table 1. Demographics.**

| Characteristics | Overall | | | <75 years old | | | ≥75 years old | | |
|---|---|---|---|---|---|---|---|---|---|
| | Low-to-moderate-intensity (n = 6220) | High-intensity (n = 1876) | P | Low-to-moderate-intensity (n = 4950) | High-intensity (n = 1527) | P | Low-to-moderate-intensity (n = 1270) | High-intensity (n = 349) | P |
| **Baseline Patients' characteristics** | | | | | | | | | |
| Age (years) | 63.1 ± 12.8 | 61.9 ± 12.7 | 0.001 | 58.7 ± 10.2 | 57.8 ± 10.0 | 0.001 | 80.1 ± 5.4 | 80.2 ± 5.2 | 0.68 |
| Male | 4484 (72.1) | 1421 (75.7) | 0.002 | 3889 (78.6) | 1249 (81.8) | 0.007 | 595 (46.9) | 172 (49.3) | 0.46 |
| Hypertension | 3185 (51.2) | 941 (50.2) | 0.43 | 2347 (47.5) | 716 (46.9) | 0.72 | 838 (66.0) | 225 (64.5) | 0.64 |
| Diabetes mellitus | 1930 (31.1) | 533 (28.4) | 0.031 | 1521 (30.8) | 420 (27.5) | 0.017 | 409 (32.2) | 113 (32.4) | 1.00 |
| Hyperlipidemia | 924 (14.9) | 562 (19.3) | <0.001 | 775 (15.7) | 317 (20.8) | <0.001 | 149 (11.7) | 45 (12.9) | 0.62 |
| Current smoker | 2571 (41.3) | 826 (44.0) | 0.041 | 2385 (48.2) | 765 (50.1) | 0.20 | 186 (14.6) | 61 (17.5) | 0.22 |
| Clinical diagnosis | | | 0.19 | | | 0.18 | | | 0.60 |
| STEMI | 3378 (54.3) | 986 (52.6) | | 2788 (56.3) | 830 (54.4) | | 590 (46.5) | 156 (44.7) | |
| NSTEMI | 2840 (45.7) | 890 (47.4) | | 2160 (43.7) | 697 (45.6) | | 680 (53.5) | 193 (55.3) | |
| CKD | 94 (1.5) | 35 (1.9) | 0.33 | 80 (1.6) | 29 (1.9) | 0.53 | 14 (1.1) | 6 (1.7) | 0.41 |
| Prior MI | 219 (3.5) | 35 (1.9) | <0.001 | 155 (3.1) | 23 (1.5) | 0.001 | 64 (5.0) | 12 (3.4) | 0.27 |
| Prior PCI | 370 (6.0) | 85 (4.5) | 0.022 | 258 (5.2) | 64 (4.2) | 0.12 | 112 (8.8) | 21 (6.0) | 0.12 |
| Prior CABG | 29 (0.5) | 3 (0.2) | 0.1 | 20 (0.4) | 1 (0.1) | 0.04 | 9 (0.7) | 2 (0.6) | 1.00 |
| Prior CVA | 427 (6.9) | 111 (5.9) | 0.16 | 300 (6.1) | 75 (4.9) | 0.10 | 127 (10.0) | 36 (10.3) | 0.94 |
| PAD | 31 (0.5) | 6 (0.3) | 0.42 | 20 (0.4) | 5 (0.3) | 0.85 | 11 (0.9) | 1 (0.3) | 0.48 |
| AF | 180 (2.9) | 21 (1.1) | <0.001 | 116 (2.3) | 14 (0.9) | 0.001 | 64 (5.0) | 7 (2.0) | 0.021 |
| LV EF, % | 53.5 ± 10.9 | 53.6 ± 10.9 | 0.94 | 54.2 ± 10.5 | 54.2 ± 10.7 | 0.94 | 51.1 ± 11.9 | 50.9 ± 11.6 | 0.84 |
| Total Cholesterol | 176 (150–204) | 182 (154–212) | <0.001 | 178 (153–206) | 184 (157–214) | <0.001 | 166 (140–195) | 168 (140–199) | 0.62 |
| Triglyceride | 100 (67–149) | 115 (77–166) | <0.001 | 104 (71–154) | 123 (81–175) | <0.001 | 84 (59–123) | 89 (63–126) | 0.09 |
| HDL Cholesterol | 40 (34–47) | 40 (34–46) | 0.15 | 40 (34–47) | 39 (34–46) | 0.25 | 41 (34–49) | 40 (34–48) | 0.43 |
| LDL Cholesterol | 112 (88–136) | 118 (93–144) | <0.001 | 114 (90–138) | 120 (96–145) | <0.001 | 105 (81–131) | 107 (81–132) | 0.52 |
| **Discharge medication** | | | | | | | | | |
| Aspirin | 6142 (98.8) | 1848 (98.7) | 0.83 | 4892 (98.8) | 1503 (98.6) | 0.44 | 1250 (98.4) | 345 (99.1) | 0.45 |
| P2Y$_{12}$ inhibitor | 6080 (98.0) | 1850 (98.9) | 0.02 | 4838 (98.1) | 1505 (98.8) | 0.06 | 1242 (97.9) | 345 (99.1) | 0.21 |
| Clopidogrel | 5392 (86.7) | 1405 (75.1) | <0.001 | 4241 (85.7) | 1108 (72.7) | <0.001 | 1151 (90.6) | 297 (85.3) | 0.006 |
| Ticagrelor | 266 (4.3) | 239 (12.7) | <0.001 | 183 (3.7) | 191 (12.5) | <0.001 | 83 (6.5) | 48 (13.8) | <0.001 |
| Prasugrel | 451 (7.3) | 213 (11.4) | <0.001 | 439 (8.9) | 213 (14.0) | <0.001 | 12 (0.9) | 0 | 0.08 |
| **Lesion and procedural characteristics** | | | | | | | | | |
| Radial access | 1248 (20.1) | 370 (19.7) | 0.20 | 960 (19.4) | 309 (20.2) | 0.35 | 288 (22.7) | 61 (17.5) | 0.015 |
| LM involved | 390 (6.3) | 122 (6.5) | 0.76 | 267 (5.4) | 98 (6.4) | 0.15 | 123 (9.7) | 24 (6.9) | 0.13 |
| pLAD involved | 2645 (42.5) | 842 (44.9) | 0.08 | 2054 (41.5) | 680 (44.5) | 0.038 | 591 (46.5) | 162 (46.4) | >0.99 |
| Disease extent | | | 0.40 | | | 0.24 | | | 0.52 |
| 1VD | 2761 (44.4) | 851 (45.4) | | 2325 (46.9) | 723 (47.4) | | 466 (36.7) | 141 (40.4) | |
| 2VD | 2056 (33.1) | 625 (33.3) | | 1620 (32.7) | 516 (33.8) | | 436 (34.3) | 109 (31.2) | |
| 3VD | 1373 (22.1) | 387 (20.6) | | 1005 (20.3) | 288 (18.9) | | 368 (29.0) | 99 (28.4) | |
| Complex PCI | 2710 (43.6) | 809 (43.1) | | 2124 (42.9) | 654 (42.8) | 0.98 | 586 (46.1) | 155 (44.4) | 0.61 |
| Total stent number | 1.63 ± 0.88 | 1.64 ± 0.89 | 0.70 | 1.61 ± 0.88 | 1.64 ± 0.90 | 0.38 | 1.68 ± 0.86 | 1.64 ± 0.87 | 0.38 |
| Mean stent diameter | 3.17 ± 0.40 | 3.16 ± 0.56 | 0.67 | 3.20 ± 0.41 | 3.19 ± 0.52 | 0.50 | 3.07 ± 0.35 | 3.07 ± 0.72 | 0.96 |
| Total stent length | 34.9 ± 21.0 | 34.6 ± 22.0 | 0.61 | 34.57 ± 21.0 | 34.65 ± 22.3 | 0.91 | 36.00 ± 21.1 | 34.26 ± 20.7 | 0.17 |

(*Continued*)

**Table 1.** (Continued)

| Characteristics | Overall | | | <75 years old | | | ≥75 years old | | |
|---|---|---|---|---|---|---|---|---|---|
| | Low-to-moderate-intensity (n = 6220) | High-intensity (n = 1876) | P | Low-to-moderate-intensity (n = 4950) | High-intensity (n = 1527) | P | Low-to-moderate-intensity (n = 1270) | High-intensity (n = 349) | P |
| IVUS use | 1352 (21.7) | 372 (19.8) | 0.08 | 1135 (22.9) | 314 (20.6) | 0.057 | 217 (17.1) | 58 (16.6) | 0.90 |

STEMI, ST-segment elevation myocardial infarction; NSTEMI, non ST-segment elevation myocardial infarction; CKD, chronic kidney disease; MI, myocardial infarction; PCI, percutaneous coronary intervention; CABG, coronary artery bypass grafting surgery; CVA, cerebrovascular attack; PAD, peripheral artery disease; AF, atrial fibrillation; LV EF, left ventricle ejection fraction; HDL, high-density lipoprotein; LDL, low-density lipoprotein; LM, left main; pLAD, proximal left anterior descending artery; VD, vessel disease; IVUS, intravascular ultrasound.

group (TVF: aHR, 0.9; 95% confidence interval [CI], 0.74–1.09; p = 0.28; CV death: aHR, 1.12; 95% CI, 0.81–1.46; p = 0.59; TV-MI: aHR, 1.08; 95% CI, 0.6–1.93; p = 0.81; TLR: aHR, 0.74; 95% CI, 0.57–0.96; p = 0.025). We also investigated the monthly incidence rates of clinical outcomes in overall population to identify a discriminating trends between acute and maintenance phase (Fig 1 in S1 Appendix). As described Fig 1 in S1 Appendix, Cochran-Armitage trend testing revealed these trends to be statistically significant in terms of TVF, CV death and TV-MI (p≤0.01), which showed relatively higher incidence rates of clinical outcomes in acute phase (<1 month) and decreased trend in maintenance phase (from 1 month to 12 months).

## Adverse clinical outcomes within acute phase (<1 month)

The incidence rates of clinical outcomes between two statin-intensity groups in the overall population, younger and elderly patients within 1 month after index PCI are shown in Table 3. Kaplan-Meier curves for TVF at 1 month were shown in Fig 2, adjusted by IPTW analysis. In overall patients, the high-intensity statin group showed poorer 1-month clinical outcomes regarding TVF, all-cause death and CV death (TVF: aHR, 1.72; 95% CI, 1.07–2.78; p = 0.026; all-cause death: aHR, 2.21; 95% CI, 1.21–4.04; p = 0.01; CV death: aHR, 2.16; 95% CI, 1.16–4.02; p = 0.015). However, after IPTW analysis, no differences were observed with statin-intensity in both younger and elderly patients (Table 3).

After subgroup analysis by aged of 75, these findings were consistent in younger patients. Meanwhile, in elderly patients, despite numerically higher event rates in the high-intensity statin group, there were no statistical significance (Table 3). No significant interactions were found between aged group and statin intensity in any of the adjusted 1-month risks of study outcomes ($P_{interaction}$ = 0.97 for TVF, $P_{interaction}$ = 0.96 for all-cause death, $P_{interaction}$ = 0.68 for CV death, $P_{interaction}$ = 0.31 for TV-MI and $P_{interaction}$ = 0.73 for TLR).

## Adverse clinical outcomes in maintenance phase (from 1 month to 12 months)

The incidence rates of clinical outcomes between two statin-intensity groups in the overall population, younger and elderly patients in maintenance phase are shown in Table 4. Kaplan-Meier curves for TVF and secondary end points in maintenance phase were shown in Fig 2 and Fig 2 in S1 Appendix, which were adjusted by IPTW analysis. In overall population, the risk of adverse clinical outcomes according to statin-intensity was not different except for TLR (TVF: aHR, 0.85; 95% CI, 0.69–1.05; p = 0.12; TLR: aHR, 0.73; 95% CI, 0.55–0.96; p = 0.022; Table 4). In younger patients, the high-intensity statin group showed significantly better clinical outcomes in terms of TVF (aHR, 0.76; 95% CI, 0.59–0.99; p = 0.038) and TLR (aHR, 0.72; 95% CI, 0.54–0.97; p = 0.032) than the low-to-moderate-intensity statin group. However,

**Table 2. Demographics after inverse probability weighting.**

| Characteristics | <75 years old | | | | ≥75 years old | | | |
|---|---|---|---|---|---|---|---|---|
| | Low-to-moderate intensity (n = 6133) | High-intensity (n = 5683) | P | SMD | Low-to-moderate intensity (n = 1548) | High-intensity (n = 1435) | P | SMD |
| **Baseline patients characteristics** | | | | | | | | |
| Age (years) | 58.5 ± 10.2 | 58.3 ± 9.8 | 0.64 | 0.015 | 80.1 ± 5.4 | 80.0 ± 5.1 | 0.76 | 0.019 |
| Male | 4878 (79.5) | 4592 (80.8) | 0.33 | 0.032 | 733 (47.4) | 691 (48.1) | 0.81 | 0.016 |
| Hypertension | 2896 (47.2) | 2689 (47.3) | 0.96 | 0.002 | 1012 (65.4) | 915 (63.8) | 0.61 | 0.034 |
| Diabetes mellitus | 1841 (30.0) | 1687 (29.7) | 0.82 | 0.007 | 505 (32.6) | 470 (32.8) | 0.97 | 0.003 |
| Hyperlipidemia | 1068 (17.4) | 1137 (20.0) | 0.039 | 0.066 | 192 (12.4) | 212 (14.8) | 0.32 | 0.068 |
| Current smoker | 2994 (48.8) | 2789 (49.1) | 0.88 | 0.005 | 235 (15.2) | 207 (14.4) | 0.74 | 0.021 |
| Clinical diagnosis | | | 0.55 | 0.019 | | | 0.99 | 0.002 |
| STEMI | 3417 (55.7) | 3111 (54.7) | | | 712 (46.0) | 659 (45.9) | | |
| NSTEMI | 2717 (44.3) | 2573 (45.3) | | | 836 (54.0) | 777 (54.1) | | |
| CKD | 99.3 (1.6) | 87.5 (1.5) | 0.84 | 0.006 | 19 (1.2) | 18 (1.3) | 0.96 | 0.003 |
| Prior MI | 156 (2.5) | 135 (2.4) | 0.77 | 0.011 | 72 (4.6) | 62 (4.4) | 0.84 | 0.014 |
| Prior PCI | 290 (4.7) | 266 (4.7) | 0.94 | 0.002 | 127 (8.2) | 116 (8.1) | 0.95 | 0.004 |
| Prior CABG | 19 (0.3) | 10 (0.2) | 0.58 | 0.027 | 11 (0.7) | 11 (0.8) | 0.90 | 0.009 |
| Prior CVA | 350 (5.7) | 323 (5.7) | 0.97 | 0.001 | 158 (10.2) | 142 (9.9) | 0.86 | 0.011 |
| PAD | 24 (0.4) | 24 (0.4) | 0.90 | 0.005 | 12 (0.8) | 10 (0.7) | 0.92 | 0.009 |
| AF | 121 (2.0) | 93 (1.6) | 0.53 | 0.025 | 70 (4.5) | 54 (3.8) | 0.65 | 0.039 |
| LV EF, % | 54.2 ± 10.5 | 54.2 ± 10.7 | 0.83 | 0.007 | 51.0 ± 11.9 | 50.7 ± 11.6 | 0.67 | 0.028 |
| **Lesion and Procedural characteristics** | | | | | | | | |
| Radial access | 1244 (20.3) | 1212 (21.3) | 0.40 | 0.026 | 343 (22.1) | 333 (23.2) | 0.71 | 0.027 |
| LM involved | 353 (5.8) | 336 (5.9) | 0.84 | 0.006 | 139 (9.0) | 118 (8.2) | 0.70 | 0.028 |
| pLAD involved | 2541 (41.4) | 2500 (44.0) | 0.11 | 0.052 | 707 (45.7) | 693 (48.3) | 0.44 | 0.051 |
| Disease extent | | | 0.44 | 0.052 | | | 0.35 | 0.12 |
| 1VD | 2939 (47.9) | 2715 (47.8) | | | 572 (37.0) | 607 (42.3) | | |
| 2VD | 1966 (32.1) | 1870 (32.9) | | | 539 (34.8) | 463 (32.2) | | |
| 3VD | 1228 (20.0) | 1099 (19.3) | | | 437 (28.2) | 366 (25.5) | | |
| Complex PCI | 2636 (43.0) | 2484 (43.7) | 0.65 | 0.015 | 705 (45.6) | 665 (46.3) | 0.82 | 0.015 |
| Total stent number | 1.62 ± 0.88 | 1.64 ± 0.91 | 0.47 | 0.025 | 1.67 ± 0.85 | 1.66 ± 0.90 | 0.87 | 0.012 |
| Mean stent diameter | 3.20 ± 0.42 | 3.20 ± 0.51 | 0.97 | 0.001 | 3.07 ± 0.35 | 3.07 ± 0.77 | 0.94 | 0.006 |
| Total stent length | 34.47 ± 20.97 | 35.20 ± 22.73 | 0.34 | 0.033 | 35.65 ± 20.66 | 34.97 ± 21.53 | 0.65 | 0.032 |
| IVUS use | 1382 (22.5) | 1279 (22.5) | 0.98 | 0.001 | 261 (16.8) | 234 (16.3) | 0.82 | 0.015 |

SMD, standardised mean difference; STEMI, ST-segment elevation myocardial infarction; NSTEMI, non ST-segment elevation myocardial infarction; CKD, chronic kidney disease; MI, myocardial infarction; PCI, percutaneous coronary intervention; CABG, coronary artery bypass grafting surgery; CVA, cerebrovascular attack; PAD, peripheral artery disease; AF, atrial fibrillation; LV EF, left ventricle ejection fraction; HDL, high-density lipoprotein; LDL, low-density lipoprotein; LM, left main; pLAD, proximal left anterior descending artery; VD, vessel disease; IVUS, intravascular ultrasound.

intriguingly, in elderly patients, the incidence rates of adverse clinical outcomes between two statin-intensity groups did not differ (TVF: aHR, 1.1; 95% CI, 0.76–1.59; p = 0.63; TLR: aHR, 0.76; 95% CI, 0.37–1.57; p = 0.46). These findings were unchanged after IPTW adjustment for differences in baseline covariates.

The forest plot of hazard ratio after IPTW analysis for adverse clinical outcomes from 1 month to 12 months was shown in Fig 3. No significant treatment interactions were detected

**Table 3. Event rates and hazard ratios for clinical outcomes in acute phase (<1 month).**

| Outcomes | Event Rates at 1 Month (n/%*) | | Crude | | Multivariate Adjusted† | | IPTW Adjusted | | |
|---|---|---|---|---|---|---|---|---|---|
| | Low-to-moderate-intensity | High-intensity | HR (95% CI) | P | HR (95% CI) | P | HR (95% CI) | P | $P_{Interaction}$ |
| **Overall** | | | | | | | | | |
| TVF | 52 (0.8)** | 25 (1.3)** | 1.6 (1.0–2.58) | 0.052 | 1.72 (1.07–2.78) | 0.026 | 1.24 (0.73–2.10) | 0.43 | 0.97 |
| All-cause death | 28 (0.5)** | 17 (0.9)** | 2.03 (1.11–3.7) | 0.022 | 2.21 (1.21–4.04) | 0.01 | 1.37 (0.70–2.70) | 0.36 | 0.96 |
| CV death | 27 (0.4)** | 16 (0.9)** | 1.98 (1.07–3.67) | 0.031 | 2.16 (1.16–4.02) | 0.015 | 1.32 (0.65–2.66) | 0.44 | 0.68 |
| TV-MI | 10 (0.2) | 6 (0.3) | 2.0 (0.73–5.49) | 0.18 | 2.07 (0.75–5.71) | 0.16 | 1.72 (0.61–4.86) | 0.31 | 0.31 |
| TLR | 23 (0.4) | 5 (0.3) | 0.72 (0.27–1.9) | 0.51 | 0.76 (0.29–2.0) | 0.58 | 0.65 (0.24–1.78) | 0.40 | 0.73 |
| **< 75 years old** | | | | | | | | | |
| TVF | 31 (0.6) | 16 (1.1) | 1.68 (0.92–3.07) | 0.09 | 1.84 (1.00–3.37) | 0.048 | 1.23 (0.63–2.38) | 0.54 | |
| All-cause death | 16 (0.3) | 10 (0.7) | 2.04 (0.92–4.49) | 0.08 | 2.34 (1.06–5.17) | 0.036 | 1.39 (0.57–3.38) | 0.47 | |
| CV death | 15 (0.3) | 10 (0.7) | 2.17 (0.98–4.84) | 0.057 | 2.49 (1.12–5.56) | 0.026 | 1.48 (0.60–3.64) | 0.39 | |
| TV-MI | 8 (0.2) | 4 (0.3) | 1.63 (0.49–5.4) | 0.43 | 1.72 (0.52–5.37) | 0.38 | 1.16 (0.35–3.90) | 0.81 | |
| TLR | 14 (0.3) | 4 (0.3) | 0.93 (0.31–2.82) | 0.90 | 0.99 (0.32–3.01) | 0.98 | 0.73 (0.23–2.29) | 0.59 | |
| **≥ 75 years old** | | | | | | | | | |
| TVF | 21 (1.7) | 9 (2.6) | 1.57 (0.72–3.44) | 0.26 | 1.57 (0.72–3.42) | 0.26 | 1.21 (0.51–2.87) | 0.66 | |
| All-cause death | 12 (0.9) | 7 (2.0) | 2.14 (0.84–5.44) | 0.11 | 2.13 (0.84–5.4) | 0.11 | 1.37 (0.46–4.04) | 0.57 | |
| CV death | 12 (0.9) | 6 (1.7) | 1.84 (0.69–4.89) | 0.22 | 1.82 (0.68–4.86) | 0.23 | 1.08 (0.34–3.37) | 0.90 | |
| TV-MI | 2 (0.2) | 2 (0.6) | 3.65 (0.51–25.9) | 0.20 | 3.63 (0.51–25.8) | 0.20 | 3.92 (0.5–30.53) | 0.19 | |
| TLR | 9 (0.7) | 1 (0.3) | 0.41 (0.05–3.2) | 0.39 | 0.41 (0.05–3.25) | 0.40 | 0.48 (0.07–3.61) | 0.48 | |

†adjusted by covariates including age, diabetes mellitus.

*Event rates were derived from the Kaplan-Meier estimates. Hazard ratio is the risk of high-intensity statin for clinical outcomes compared with that of less intensive statin.

**P value by log-rank test was less than 0.05.

HR, hazard ratio; TVF, target vessel failure; CV, cardiovascular; TV-MI, target-vessel myocardial infarction; TLR, target lesion revascularization.

in subgroups defined by aged of 75 ($P_{interaction}$ = 0.18 for TVF, $P_{interaction}$ = 0.35 for all-cause death, $P_{interaction}$ = 0.15 for CV death, and $P_{interaction}$ = 0.76 for TLR).

## Sensitivity analysis after PS matching

After PS matching between high- and low-to-moderate-intensity statin groups, there were 1,398 and 305 matched pairs in younger and elderly patients, respectively. There were no other significant differences for any of the covariates between two statin-intensity groups, except for some covariates (hypertension, hyperlipidemia, clinical diagnosis, atrial fibrillation, baseline value of triglyceride and use of IVUS) in younger patients (Table 2 in S1 Appendix).

The adverse clinical outcomes in this sub-cohort and in each group stratified by statin-intensity are shown in Table 3 in S1 Appendix and Fig 3 in S1 Appendix. These sensitivity analyses revealed consistent findings with the relative effect of high- and low-to-moderate-intensity statin according to aged groups, in which the better clinical outcomes of high-intensity statin group was more prominent in younger patients.

## Subgroup analysis for clinical impact of statin-intensity among patients with a maintained intensity of statin

In order to further support the result of our study, we performed subgroup analysis with patients who continued the same intensity of statin therapy. Because adherence and tolerance are important clinical aspects of medical treatment that affect clinical outcomes [20], we

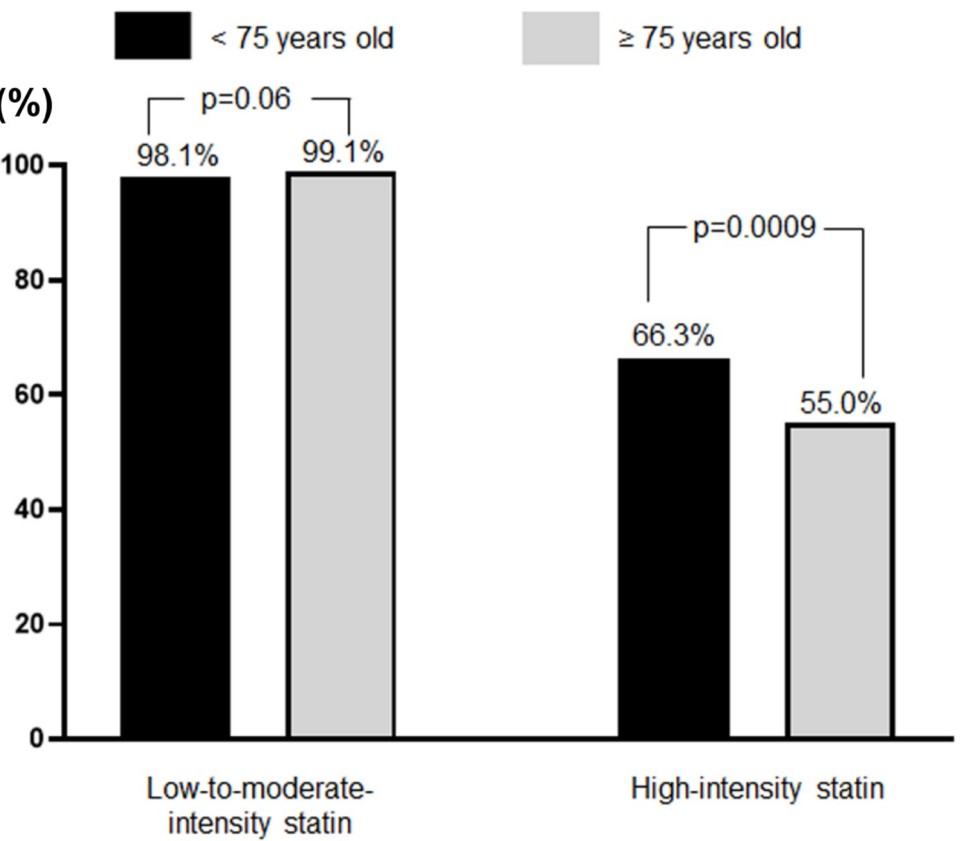

**Fig 2. Adjusted Kaplan-Meier curves for the primary end point according to statin-intensity in younger and elderly patients using inverse probability weighting.**

investigated the maintenance proportion of the same intensity of statin during 1 year after index AMI in total of 6,566 patients. In high-intensity statin group, the proportion of patients continuing the same statin-intensity was significantly lower in elderly patients than that of younger patients (55.0% vs. 66.3%, p = 0.0009), while this phenomenon was not observed in low-to-moderate-intensity statin group (99.1% vs. 98.1%, p = 0.06) (Fig 4).

Table 5 in S1 Appendix shows the results of subgroup analysis, patients who maintained high-intensity statin showed significantly better clinical outcome in terms of TVF in overall and younger patients (overall: aHR 0.64, 95% CI 0.46–0.9, p = 0.01; younger patients: aHR 0.63, 95% CI 0.44–0.9, p = 0.012). However, in elderly patients, the incidence rates of adverse clinical outcomes did not differ between continuous use of high- and low-to-moderate-intensity statin (aHR 0.7, 95% CI 0.28–1.77, p = 0.46). These findings were consistent with the main results of our study. However, the incidence rate of TVF was numerically lower in high-intensity statin group compared to low-to-moderate-intensity statin group. To overcome the small number of event rates, we extended a study period to 24 months, which also showed a consistent result (Table 5 in S1 Appendix).

### Independent Predictors of TVF

As shown in Table 4 in S1 Appendix, in younger patients, high-intensity statin revealed to be an independently protective effect for TVF in maintenance phase (HR, 0.77; 95% CI, 0.59–0.99; p = 0.04), which was not in elderly patients.

**Table 4. Event rates and hazard ratios for clinical outcomes in maintenance phase (from 1 month to 12 months).**

| Outcomes | Event Rates at 1–12 Month (n/%*) | | Crude | | Multivariate Adjusted† | | IPTW Adjusted | | |
|---|---|---|---|---|---|---|---|---|---|
| | Low-to-moderate-intensity | High-intensity | HR (95% CI) | P | HR (95% CI) | P | HR (95% CI) | P | P_Interaction |
| **Overall** | | | | | | | | | |
| TVF | 460 (7.7) | 117 (6.6) | 0.86 (0.7–1.05) | 0.14 | 0.85 (0.69–1.05) | 0.12 | 0.85 (0.68–1.06) | 0.15 | 0.17 |
| All-cause death | 220 (3.7) | 71 (4.0) | 1.1 (0.84–1.43) | 0.50 | 1.15 (0.87–1.52) | 0.33 | 1.17 (0.86–1.57) | 0.32 | 0.30 |
| CV death | 164 (2.7) | 53 (3.0) | 1.1 (0.8–1.49) | 0.56 | 1.1 (0.79–1.53) | 0.58 | 1.06 (0.75–1.49) | 0.76 | 0.28 |
| TV-MI | 38 (0.6) | 10 (0.6) | 0.89 (0.45–1.79) | 0.75 | 0.83 (0.4–1.73) | 0.62 | 0.84 (0.37–1.92) | 0.69 | NA |
| TLR | 295 (5.0)** | 64 (3.7)** | 0.73 (0.56–0.96) | 0.024 | 0.73 (0.55–0.96) | 0.022 | 0.72 (0.54–0.97) | 0.033 | 0.76 |
| **< 75 years old** | | | | | | | | | |
| TVF | 329 (6.9) | 79 (5.5) | 0.79 (0.62–1.01) | 0.059 | 0.76 (0.59–0.99) | 0.038 | 0.75 (0.57–0.99) | 0.044 | |
| All-cause death | 107 (2.2) | 32 (2.2) | 0.99 (0.67–1.47) | 0.95 | 0.97 (0.63–1.49) | 0.89 | 0.96 (0.61–1.52) | 0.86 | |
| CV death | 76 (1.6) | 22 (1.5) | 0.96 (0.59–1.54) | 0.85 | 0.85 (0.5–1.46) | 0.56 | 0.82 (0.47–1.43) | 0.48 | |
| TV-MI | 27 (0.6) | 10 (0.7) | 1.22 (0.59–2.53) | 0.59 | 1.11 (0.52–2.39) | 0.79 | 1.19 (0.51–2.76) | 0.69 | |
| TLR | 250 (5.3)** | 55 (3.9)** | 0.72 (0.54–0.97) | 0.029 | 0.71 (0.53–0.96) | 0.025 | 0.70 (0.51–0.97) | 0.033 | |
| **≥ 75 years old** | | | | | | | | | |
| TVF | 131 (10.8) | 38 (11.7) | 1.1 (0.76–1.58) | 0.61 | 1.1 (0.76–1.59) | 0.63 | 1.06 (0.72–1.57) | 0.76 | |
| All-cause death | 113 (9.2) | 39 (11.8) | 1.32 (0.91–1.89) | 0.14 | 1.35 (0.92–1.97) | 0.12 | 1.33 (0.90–1.97) | 0.15 | |
| CV death | 88 (7.2) | 31 (9.5) | 1.34 (0.89–2.02) | 0.16 | 1.34 (0.88–2.05) | 0.18 | 1.24 (0.80–1.92) | 0.35 | |
| TV-MI | 11 (0.9) | 0 | NA | | NA | | NA | | |
| TLR | 45 (3.9) | 9 (2.9) | 0.76 (0.37–1.55) | 0.45 | 0.76 (0.37–1.57) | 0.46 | 0.81 (0.37–1.75) | 0.59 | |

†adjusted by covariates including age, diabetes mellitus.

*Event rates were derived from the Kaplan-Meier estimates. Hazard ratio is the risk of high-intensity statin for clinical outcomes compared with that of less intensive statin.

**P value by log-rank test was less than 0.05.

HR, hazard ratio; TVF, target vessel failure; CV, cardiovascular; TV-MI, target-vessel myocardial infarction; TLR, target lesion revascularization.

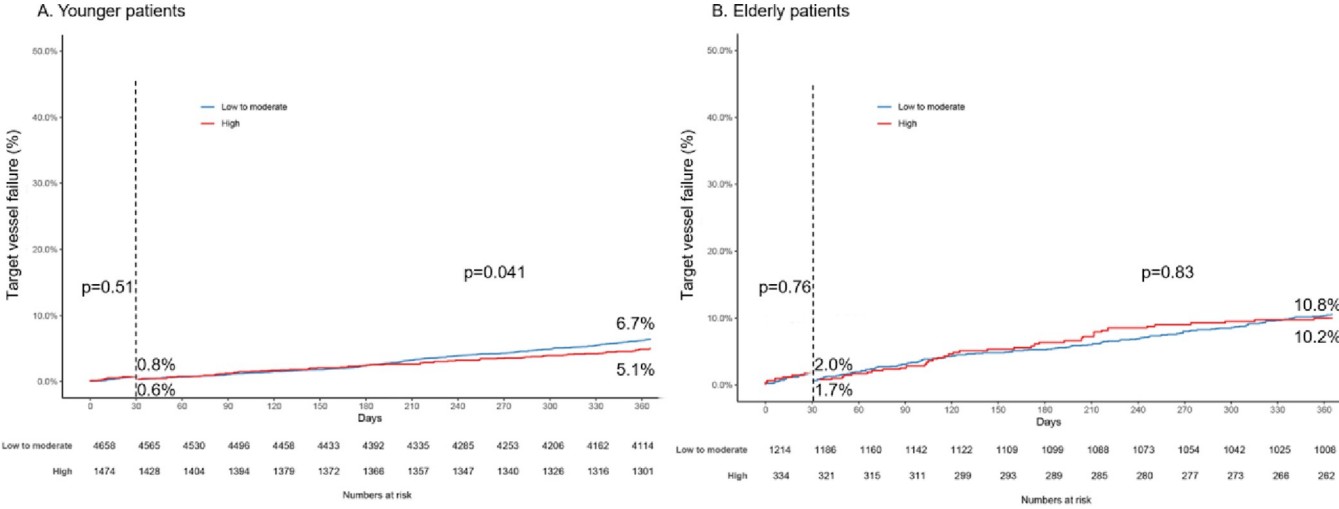

**Fig 3. Effects of statin intensity on clinical outcomes in maintenance phase after IPTW analysis, subdivided by aged of 75.** *derived from unmatched population. **aHR, adjusted hazard ratio for high-intensity statin treatment compared with less intensive strategy after IPTW analysis. TVF, target-vessel failure; TLR, target lesion revascularization; HR, hazard ratio; NA, not applicable.

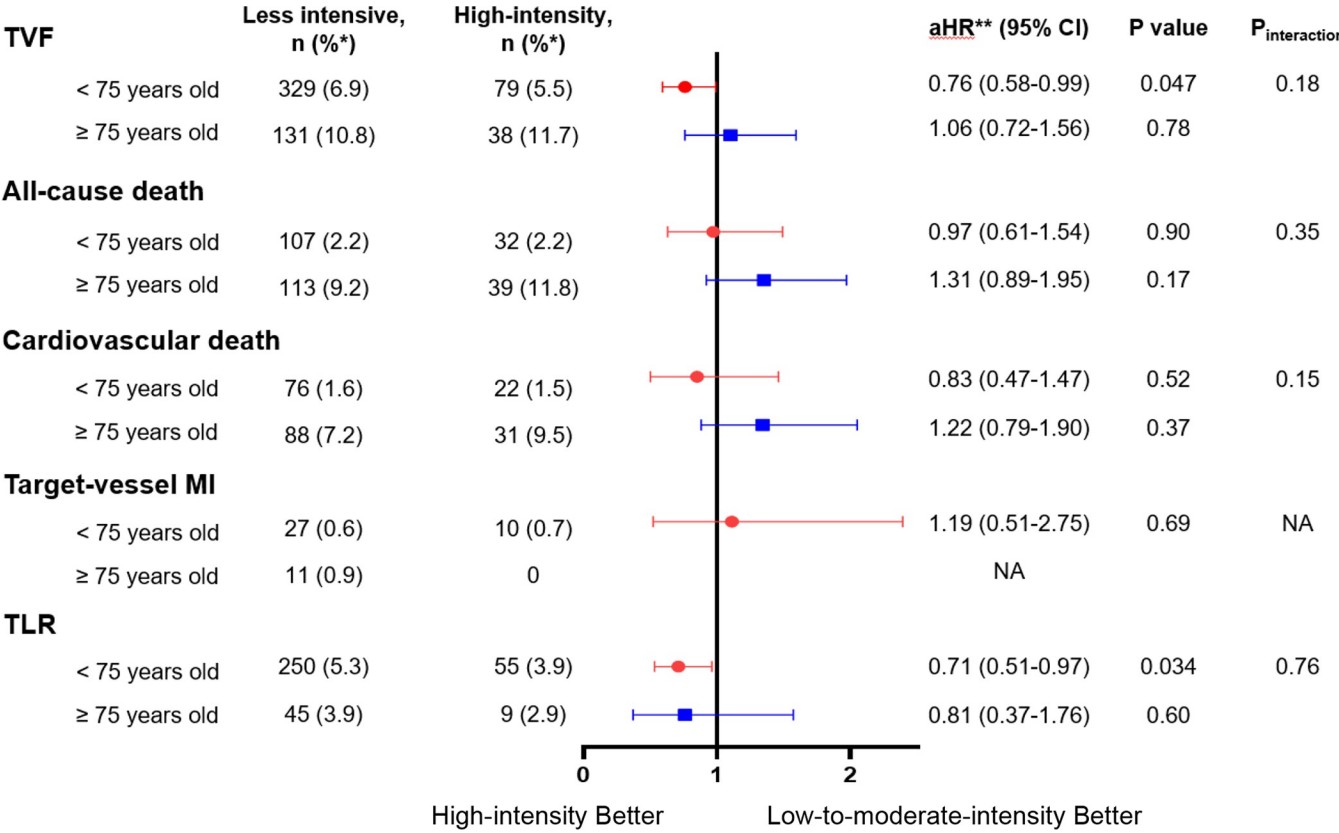

**Fig 4. Age differences on maintenance proportion of statin-intensity.**

## Discussion

The major findings from the present study are as follows: 1) although current guideline recommends high-intensity statin in AMI patients, low-to-moderate-intensity statins were prescribed 3 times more than high-intensity statins in daily clinical practice; 2) in high-intensity statin group, the proportion of patients continuing the same statin-intensity was significantly lower in elderly patients than that of younger patients; 3) in maintenance phase, high-intensity statin group showed the better clinical outcomes in terms of TVF, mainly by TLR than less-intensity group among younger patients; 4) in contrast, the incidence rates of adverse clinical events between high- and low-to-moderate-intensity statin were not statistically different in elderly patients.

Because the Corea-AMI registry is large, prospective, multicenter registry enrolled 10,719 AMI patients underwent PCI, we believe that our findings may reflect real-world clinical practice for secondary prevention regarding the statin-intensity. Indeed, we enrolled consecutive AMI patients with end-stage renal disease, heart failure or cardiogenic shock. Interestingly, in our data, less-intensive statins rather than high-intensity statins were prescribed 3 times more often regardless of age group (4,950 of 6,477 [76.4%] in younger patients and 1,270 of 1,619 [78.4%] in elderly patients; p = 0.09). Similar with our finding, several previous studies also have shown that high-intensity statin was not frequently used for ASCVD patients in the real-world clinical practice, in which only in 15–29.6% of patients received high-intensity statin [20–22].

Moreover, de-escalation of high-intensity statin was frequently observed in elderly patients during 1 year. Likewise, in the Patient and Provider Assessment of Lipid Management (PALM) registry, elderly patients were much more likely to receive a moderate-intensity statin rather than a high-intensity statin for secondary prevention in real-world situation [23]. The plausible explanations of this situation are paucity of evidence using high-intensity statin in elderly patients, concern about side-effect of high-intensity statin and poor compliance in elderly patients. Actually, 2019 ESC/EAS guideline mentions that statin is started at a low dose and titrated upwards if there is significant renal impairment or the potential for drug interactions, which is more likely to be observed in elderly patients [11].

In our data, through the 1-month landmark analysis, it was shown that the risk of the early adverse clinical outcomes was significantly higher in high-intensity statin than in less-intensive statin group. This may be caused by patients' unstable condition or incomplete procedural factors during acute phase of AMI, which reflected that high-intensity statin was more likely to be prescribed in higher risk patients who were expected to have a poor prognosis.

In addition, we demonstrated that high-intensity statin showed better clinical outcomes than low-to-moderate-intensity statin in younger patients during maintenance phase, while the clinical outcomes between two statin-intensity group was similar in elderly patients. There have been several studies regarding clinical efficacy of low-to-moderate-intensity statin for secondary prevention in elderly patients. Looking at the result of Pravastatin or Atorvastatin Evaluation and Infection Therapy-Thrombolysis in Myocardial Infarction 22 (PROVE IT-TIMI 22) Trial [24], high-intensity atorvastatin was not superior to moderate-intensity pravastatin among patients aged $\geq$ 65 years. IDEAL (Incremental DEcrease through Aggressive Lipid Lowering) Study randomly assigned 8,888 patients age $\leq$ 80 years with previous myocardial infarction to receive either intensive therapy with atorvastatin 80 mg/day or the standard statin therapy (simvastatin 20 to 40 mg/day) to compare the ability of these regimens to lower cardiovascular risk. In the subgroup analysis, significant reductions in primary and secondary endpoints were observed only in patients <65 years of age compared to 65 to 80 years of age [25]. Kwak A. et al. [26] reported that moderate-intensity statin can be more protective against ischemic cardiovascular and cerebrovascular events compared with high-intensity statin in patients aged 75 years and older. Foody J.M. et al. [27] showed that statin therapy was associated with a significant reduction in all-cause mortality in patients younger than 80, but not in patients aged 80 and older. Although some analyses reported an association between high-intensity statin use and a survival benefit in older adults, the study population was heterogenous while our study population consisted of only AMI patients [28]. Even about 60% of CAD patients were included in the study population, no information of severity or revascularization of coronary lesions was available in this study [28]. Besides, in the meta-analysis by Cholesterol Treatment Trialists' Collaboration, 26 trials were based on analysis between statin therapy versus control, not comparison of statin-intensity [29]. Thus, we need well-powered large scale randomized studies regarding this topic.

Lower efficacy of high-intensity statin than expected in patients over 75 years of age may be explained by lower endogenous cholesterol synthesis and higher cholesterol absorption in elderly subjects. In the DEBATE study of home-dwelling elderly patients, low cholesterol absorption was associated with fewer cardiovascular events and better survival, while increased cholesterol absorption was associated with increased cardiovascular mortality [30]. Also, in subgroup analysis of IMPROVE-IT, Bach RG et al. found that elderly patients did benefit in particular from the addition of ezetimibe to simvastatin. Compared with younger patients, the absolute risk reduction for the primary end point was substantially greater for patients 75 years or older [31]. These findings raise the hypothesis that cholesterol metabolism changes during lifetime: whereas midlife is characterized by higher endogenous synthesis rates and

lower cholesterol absorption rates, cholesterol synthesis deteriorates with increasing age. Thus, we think that the decreased importance of endogenous cholesterol synthesis and the increased role of cholesterol absorption with increasing age may be the potential explanation of our findings. Taken together, in elderly patients who are not tolerant to high-intensity statin, we should consider additional optional treatment with statin such as ezetimibe to achieve a target of LDL-cholesterol level.

Our study has several limitations listed as follows. 1) This was a non-randomized, observational registry study with inherent methodological limitations; thus, its overall findings must be considered as hypothesis generating only. However, even though there is an inherent limitation, we could demonstrate the efficacy of high-intensity statin for secondary prevention in younger patients. This finding is well in line with the evidence provided by the current guidelines [11,12]. 2) Our findings are subject to selection bias because the treatment choice was left to the physician. However, to minimize the bias, we performed rigorous adjustments using a multivariable Cox proportional hazard regression model, PS matching and subgroup analysis with patients who were continuing the same statin-intensity during 1 year. Nevertheless, hidden bias may still remain because of the influence of unmeasured confounding factors. Therefore, our findings should be considered with caution until additional large clinical studies with long-term follow-ups replicate our findings. 3) Owing to the relatively limited number, some result of this study might not be sufficiently powered to compare between statin-intensity groups. Especially in elderly patients, high-intensity statin showed numerically higher incidence rates of TVF without statistical significance in maintenance phase. As mentioned above, we additionally performed subgroup analysis with patients who were continuing the same statin-intensity during 1 year. In this subgroup of elderly patients, the incidence rate of TVF was numerically lower compared to low-to-moderate-intensity statin group, which was consistent throughout extended study period of 24 months. 4) No data was available regarding side effects of statin and prescription rates of ezetimibe or PCSK9 inhibitor. Especially, PCSK9 inhibitor was not widely used in the study period due to governmental insurance coverage. 5) The study enrolled only South Korean population. Whether the results of the current study can be extrapolated to other ethnicities is unknown. Caution is needed in extrapolating these results outside of South Korea.

Nevertheless, we would like to emphasize that this study is the first research to answer the clinical impact of statin-intensity in elderly patients using large AMI registry.

## Conclusion

In this AMI cohort underwent PCI, high-intensity statin showed better clinical outcomes at 1-year than low-to-moderate intensity statin in younger patients. Meanwhile, in elderly patients, Meanwhile, the incidence rates of adverse clinical events were not statistically different between high- and low-to-moderate-intensity statin in elderly patients. Further randomized study with large elderly population is warranted.

## Supporting information

**S1 Data.**
(XLSX)

**S1 Appendix.**
(DOCX)

## Acknowledgments

There was no industry involvement in the design, conduct, or analysis of the study.

## Author Contributions

**Conceptualization:** Dae-Won Kim, Mahn-Won Park.

**Data curation:** Kyusup Lee, Myunhee Lee, Dae-Won Kim, Sungmin Lim, Eun Ho Choo, Chan Joon Kim, Chul Soo Park, Hee Yeol Kim, Ki-Dong Yoo, Doo Soo Jeon, Kiyuk Chang, Wook-Sung Chung, Min Chul Kim, Myung Ho Jeong, Youngkeun Ahn, Mahn-Won Park.

**Formal analysis:** Kyusup Lee, Myunhee Lee, Jinseob Kim, Chul Soo Park, Mahn-Won Park.

**Funding acquisition:** Kiyuk Chang.

**Investigation:** Kyusup Lee, Myunhee Lee, Sungmin Lim, Eun Ho Choo, Chan Joon Kim, Chul Soo Park, Kiyuk Chang, Ho Joong Youn, Wook-Sung Chung, Jongbum Kwon, Mahn-Won Park.

**Methodology:** Kyusup Lee, Myunhee Lee, Dae-Won Kim, Sungmin Lim, Chan Joon Kim, Ki-Dong Yoo, Doo Soo Jeon, Kiyuk Chang, Ho Joong Youn, Wook-Sung Chung, Min Chul Kim, Mahn-Won Park.

**Project administration:** Kyusup Lee.

**Resources:** Dae-Won Kim, Jinseob Kim, Eun Ho Choo, Hee Yeol Kim, Doo Soo Jeon, Kiyuk Chang, Wook-Sung Chung, Min Chul Kim, Myung Ho Jeong, Youngkeun Ahn.

**Software:** Jinseob Kim.

**Supervision:** Hee Yeol Kim, Ki-Dong Yoo, Ho Joong Youn, Wook-Sung Chung, Jongbum Kwon, Mahn-Won Park.

**Validation:** Kyusup Lee, Jinseob Kim, Chan Joon Kim, Chul Soo Park, Kiyuk Chang, Ho Joong Youn, Wook-Sung Chung, Jongbum Kwon, Mahn-Won Park.

**Visualization:** Kyusup Lee, Jinseob Kim, Mahn-Won Park.

**Writing – original draft:** Kyusup Lee.

**Writing – review & editing:** Mahn-Won Park.

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
