## [Decision Letter · Decision Letter 0]

2 Aug 2021

PONE-D-21-14585

Clinical Impact of Statin Intensity According to Age in Patients with Acute Myocardial Infarction

PLOS ONE

Dear Dr. Park,

Thank you for submitting your manuscript to PLOS ONE. After careful consideration, we feel that it has merit but does not fully meet PLOS ONE’s publication criteria as it currently stands. Therefore, we invite you to submit a revised version of the manuscript that addresses the points raised during the review process.

We look forward to receiving your revised manuscript.

Kind regards,

R. Jay Widmer

Academic Editor

PLOS ONE

Additional Editor Comments (if provided):

The reviewers were generally favorable, yet had severe concerns regarding some of the methods and variables. Please pay careful attention to each of these comments and address each individually in your response and revisions.

Reviewers' comments:

Reviewer's Responses to Questions

**Comments to the Author**

1. Is the manuscript technically sound, and do the data support the conclusions?

Reviewer #1: No

Reviewer #2: Partly

2. Has the statistical analysis been performed appropriately and rigorously? 

Reviewer #1: Yes

Reviewer #2: Yes

3. Have the authors made all data underlying the findings in their manuscript fully available?

Reviewer #1: No

Reviewer #2: No

4. Is the manuscript presented in an intelligible fashion and written in standard English?

Reviewer #1: No

Reviewer #2: Yes

5. Review Comments to the Author

Reviewer #1: The manuscript by Lee et al examines more than 10,000 patients with AMI and the clinical impact of statin use. Specifically, they demonstrate better clinical outcomes at 1-year in patients <75 treated with a high-intensity statin compared with a low-to-moderate-intensity statin. There are some concerns that should be addressed by the authors.

1. A major emphasis of the manuscript is that "less-intensive statin [sic] showed feasible and acceptable clinical outcomes compared to high-intensity statin in elderly patients." Yet, the authors themselves acknowledge that the findings are subject to selection bias from the physicians making treatment decisions and the limited sample size of older adult patients included in the propensity matched cohort (just 308 older adults matched) that prevents the comparison between statin-intensity groups in elderly patients. Thus, I don't think the authors can make any firm conclusions from that particular comparison and the current wording of both the abstract and conclusion is a bit misleading.

2. The finding that high-intensity statin use does not benefit older adults runs counter to larger prior analyses that support an association between high-intensity statin use and a survival advantage in older adults. This is more reason to temper the conclusions around statin intensity in older patients with MI.

Rodriguez F, Maron DJ, Knowles JW, Virani SS, Lin S and Heidenreich PA. Association Between Intensity of Statin Therapy and Mortality in Patients With Atherosclerotic Cardiovascular Disease. JAMA Cardiol. 2017;2:47-54

CTT. Efficacy and safety of statin therapy in older people: a meta-analysis of individual participant data from 28 randomised controlled trials. Lancet. 2019;393:407-415.

3. The higher risk of clinical events within the first month in the high-intensity statin treated group points to the likely selection bias inherent in this sort of study design. Further supports the likelihood that cannot ascribe the clinical events observed to a treatment effect of the statin.

4. We have seen from PROSPER, PALM and other studies that statins are generally quite well tolerated in older adults. While there are concerns about polypharmacy, frailty and drug interactions, the benefits likely outweigh the risks of high-intensity statin treatment in older adults with known ASCVD.

Reviewer #2: 1. The change of the amount of statin during follow-up due to side effects or lower the cholesterol level further is very common but not showed in this study. Cholesterol level at follow-up and use of rate of ezetimibe or PCSK9 did not show. Those variables can affect the clinical outcomes.

2. In this study did not showed combined outcomes between 0 to 1-month and 1 to 12-month together in manuscript, which seems to be more important.

3. This study compared TVF in high dose statin group of >75 was higher than low to moderate dose, even statistically insignificant. Explanations are needed.

4. The positive results of this study are marginal, which is hard to lead to the conclusion.

5. In discussion, page 14 line 20, 1-month seems to be change into 1-year.

6. PLOS authors have the option to publish the peer review history of their article (what does this mean?). If published, this will include your full peer review and any attached files.

Reviewer #1: No

Reviewer #2: No

---

## [Author Response · Author response to Decision Letter 0]

7 Oct 2021

Revision Response Letter

Ref. No: PONE-D-21-14585

Title: Clinical Impact of Statin Intensity According to Age in Patients with Acute Myocardial Infarction.

Dear Dr. Emily Chenette

Academic Editor of PLOS ONE

We thank you, the reviewers and editorial staff for the constructive and valuable comments regarding our manuscript. We now provide a revised manuscript for consideration of publication in PLOS ONE. We have carefully considered each comment and attached a point-by-point-response to the reviewers’ comments. We hope that the paper will be of substantial interest to the readership of PLOS ONE. All authors concur with the submission and the material has not been published previously or is not under consideration for publication elsewhere.

On behalf of the authors and with best regards

Mahn-Won Park, M.D., Ph.D., Associate Professor

Department of Cardiology, Daejeon St. Mary’s Hospital, College of Medicine, 

The Catholic University of Korea

64, Daeheung-ro, Jung-gu, Daejeon, 34943, Korea

Phone: +82-10-3800-2817, E-mail: pmw6193@catholic.ac.kr 

We thank to the editor and reviewers for their time and input, and their precious comments.

Answer : We changed the title and subtitle with capitalization only the first word.

Modified text : Line 3

Clinical impact of statin intensity according to age in patients with 

acute myocardial infarction

Short title: Statin intensity in elderly patients

Answer : We changed the author names without titles (MD, PhD).

 Modified text : Line 8

Kyusup Lee1,2, Myunhee Lee1,2, Dae-Won Kim1,2, Jinseob Kim3, Sungmin Lim1,4,

Eun Ho Choo1,9, Chan Joon Kim1,4, Chul Soo Park1,5, Hee Yeol Kim1,6, Ki-Dong Yoo1,7, 

Doo Soo Jeon1,8, Kiyuk Chang1,9, Ho Joong Youn1,9, Wook-Sung Chung1,9, Min Chul Kim10,

Myung Ho Jeong10, Youngkeun Ahn10, Jongbum Kwon11, and Mahn-Won Park1,2.

Answer : We added the ‘Data Avilability statement’ in the cover letter.

Data Availability: The data cannot be made public due to the protection of the patients’ personal information. Data are available from the Department of Cardiology in the Daejeon St. Mary's Hospital (contact via pmw6193@catholic.ac.kr - the person in charge of data management at the Department of Cardiology in the Daejeon St. Mary's Hospital) for researchers who meet the criteria for access to confidential data.

 Answer: Mahn-Won Park, ORCID Id: 0000-0001-5293-8461. Thanks.

 

Reviewers' comments

Reviewer #1: The manuscript by Lee et al examines more than 10,000 patients with AMI and the clinical impact of statin use. Specifically, they demonstrate better clinical outcomes at 1-year in patients <75 treated with a high-intensity statin compared with a low-to-moderate-intensity statin. There are some concerns that should be addressed by the authors.

1. A major emphasis of the manuscript is that "less-intensive statin [sic] showed feasible and acceptable clinical outcomes compared to high-intensity statin in elderly patients." Yet, the authors themselves acknowledge that the findings are subject to selection bias from the physicians making treatment decisions and the limited sample size of older adult patients included in the propensity matched cohort (just 308 older adults matched) that prevents the comparison between statin-intensity groups in elderly patients. Thus, I don't think the authors can make any firm conclusions from that particular comparison and the current wording of both the abstract and conclusion is a bit misleading.

Answer : Thanks for your valuable comment. As you pointed out, our findings are subject to selection bias and confounding factors because of its non-randomized, observational design. This is inherent limitation of real-world observational studies. Thus, to minimize these biases, we performed rigorous adjustments using a multivariable Cox proportional hazard regression model and PS matching, but hidden bias may still remain because of the influence of unmeasured confounding factors. We already mentioned this in the ‘Limitation’ section in the main manuscript. Therefore, because these factors may make it difficult to draw definite conclusions on the current issues, our findings should be considered with caution until additional large clinical studies with long-term follow-ups replicate our findings. However, it is also known that randomized controlled trial may not reflect real world clinical situation due to its strict inclusion criteria. In this regard, we would like to emphasize the strength of our study. First, our study is the first and the largest study investigating the clinical impact of statin intenstiy according to age in homogenous AMI population, even though the sample size of elderly patients is relatively small. Second, we only included AMI patients underwent PCI with DES. Currently, PCI with DES is regarded as the standard of treatment in AMI patients. Thus, we think that our findings can reflect current clinical practice in this population.

In addition, to respond your comments and to further support the results of our study, we additionally performed subgroup analysis with patients who were continuing the same intensity of statin. Because adherence and tolerance are important clinical aspects of medical treatment that affect clinical outcomes, we investigated the maintenance proportion of the same intensity of statin during 1 year after index AMI in total of 6,566 patients. In patients who received high-intensity statin at discharge, the proportion of patients who maintained the same intensity of statin was significantly lower in elderly patients than in younger patients (55.0% vs. 66.3%, p=0.0009), which was not shown in low-to-moderate-intensity statin group (99.1% vs. 98.1%, p=0.06). In other words, de-escalation of statin-intensity was frequently ocurred in elderly patients treated with high-intensity statin.

In overall and younger patients, patients who maintained high-intensity statin showed significantly better clinical outcome in terms of TVF (overall: HR 0.64, 95% CI 0.46-0.9, p=0.01; younger patients: HR 0.63, 95% CI 0.44-0.9, p=0.012). However, in elderly patients, the efficacy for reducing adverse clinical outcomes did not differ between continuous use of high- and low-to-moderate-intensity statin (HR 0.7, 95% CI 0.28-1.77, p=0.46). These findings were consistent with the main result of our study. However, the incidence rate of TVF was numerically lower in high-intensity statin group compared to low-to-moderate-intensity statin group. To overcome the small number of event rates, we extended a study period to 24 months, which also showed a consistent result (Appendix Table 5).

Finally, to respond your comment, we tempered the conclusion as you recommended and added a text as below. We ask for your understanding. Thanks a lot.

Original text : Line 66, abstract

Meanwhile, in elderly patients, high-intensity statin showed similar clinical outcomes compared to low-to-moderate-intensity statin, which meant less-intensive statin showed feasible and acceptable clinical outcomes compared to high-intensity statin in elderly patients.

Modified text :

Meanwhile, in elderly patients, there is no difference in clinical outcomes according to statin intensity. Further randomized study with large population of elderly is warranted.

Original text : Line 342, conclusion

In elderly patients, less intensive statin showed feasible and acceptable clinical outcomes compared to high-intensity statin for secondary prevention considering compliance, concern about side effect and drug interaction.

Modified text : 

Meanwhile, in elderly patients, clinical outcomes were not different between high-intenstiy and low-to-moderate-intensity statin. Further randomized study with large population of elderly is warranted.

Original text : Line 261, results

Modified text : 

Subgroup analysis for clinical impact of statin-intensity among patients continuing statin-intensity

In order to further support the results of our study, we performed subgroup analysis with patients who were continuing the same intensity of statin. Because adherence and tolerance are important clinical aspects of medical treatment that affect clinical outcomes, we investigated the maintenance proportion of the same intensity of statin during 1 year after index AMI in total of 6,566 patients. In patients who received high-intensity statin at discharge, the proportion of patients who maintained the same intensity of statin was significantly lower in elderly patients than that of younger patients (55.0% vs. 66.3%, p=0.0009), which was not shown in low-to-moderate-intensity statin group (99.1% vs. 98.1%, p=0.06) (Figure 2). In other words, de-escalation of statin-intensity was frequently occurred in elderly patients treated with high-intensity statin.

In overall and younger patients, patients who maintained high-intensity statin showed significantly better clinical outcome in terms of TVF (overall: HR 0.64, 95% CI 0.46-0.9, p=0.01; younger patients: HR 0.63, 95% CI 0.44-0.9, p=0.012). However, in elderly patients, the efficacy for reducing adverse clinical outcomes did not differ between continuous use of high- and low-to-moderate-intensity statin (HR 0.7, 95% CI 0.28-1.77, p=0.46). These findings were consistent with the main results of our study. However, the incidence rate of TVF was numerically lower in high-intensity statin group compared to low-to-moderate-intensity statin group. To overcome the small number of event rates, we extended a study period to 24 months, which showed a consistent result (Appendix Table 5).

Original text : Line 328

Although propensity-score analyses were performed to adjust for potential selection bias, the unmeasured confounders might have affected the results.

Modified text :

However, to minimize the bias, we performed rigorous adjustments using a multivariable Cox proportional hazard regression model and PS matching and subgroup analysis with patients who were continuing the same statin-intensity during 1 year. Nevertheless, hidden bias may still remain because of the influence of unmeasured confounding factors. Therefore, our findings should be considered with caution until additional large clinical studies with long-term follow-ups replicate our findings.

Figure 2. Age differences on maintenance proportion of statin-intensity.

Appendix Table 5. Subgroup analysis of event rates and hazard ratios for target-vessel failure among patients with continuing statin-intensity at 1 year.

2. The finding that high-intensity statin use does not benefit older adults runs counter to larger prior analyses that support an association between high-intensity statin use and a survival advantage in older adults. This is more reason to temper the conclusions around statin intensity in older patients with MI.

Rodriguez F, Maron DJ, Knowles JW, Virani SS, Lin S and Heidenreich PA. Association Between Intensity of Statin Therapy and Mortality in Patients With Atherosclerotic Cardiovascular Disease. JAMA Cardiol. 2017;2:47-54

CTT. Efficacy and safety of statin therapy in older people: a meta-analysis of individual participant data from 28 randomised controlled trials. Lancet. 2019;393:407-415.

Answer : We appreciate your valuable and attentive advice with references. With response to your comment No #1, We already tempered the conclusion as mentioned above and added those references that you suggested in the ‘Discussion’ session. 

In general, elderly individuals have a shorter life expectancy and more comorbidities than younger people, so statins may have fewer benefits in this population. This is why in elderly people, the benefit and disadvantages of the treatment with statins should be put in balance, especially in those receiving high-intensity of statins. However, it is true that the amount of information is not enough regarding the effects of high-intensity statin therapy for elderly patients with ASCVD. For this reason, previous version of ACC/AHA cholesterol guideline recommended moderate-intensity statin therapy for patients older than 75 years of age. [1] The current guideline also recommends that in patients older than 75 years of age with clinical ASCVD, it is reasonable to initiate moderate- or high-intensity statin therapy after evaluation of the potential for ASCVD risk reduction, adverse effects, and drug–drug interactions, as well as patient frailty and patient preferences (COR IIa) [2]

There are several previous studies which showed the similar clinical efficacy of less-intensity statin compared to high-intensity statin in CAD population. [3,4] IDEAL (Incremental DEcrease through Aggressive Lipid Lowering) Study randomly assigned 8,888 patients age≤ 80 years with previous myocardial infarction to receive either intensive therapy with atorvastatin 80 mg/day or the standard statin therapy at the time (simvastatin 20 to 40 mg/day) to compare the ability of these regimens to lower cardiovascular risk. In the subgroup analysis, significant reductions in primary and secondary endpoints were observed only in patients <65 years of age compared to 65 to 80 years of age (please see below figure).[4]

Figure. Intensive atorvastatin therapy reduced event rates in both younger and older patients.

Especially, there has been various studies enrolling Asian population which have shown similar clinical efficacy of less-intensive statin compared to high-intensity statin. Our group reported that the efficacy of moderate-intensity statin therapy was comparable to that of high-intensity statin therapy in terms of improved clinical outcomes using national health insurance claims (moderate intensity group, n=23,863, high intensity group, n=9,073) data in South Korea after propensity matching analysis.[5]

In addition, a pharmacokinetic study also indicated that the greater effect of statins could be due in part to the difference in statin pharmacokinetics between East Asian and Western patients.[6] In prior studies with East Asian populations, lower-dose statin therapy showed similar therapeutic effects to those observed in Western populations using higher-dose statin therapy [7,8]. In addition, serial intravascular ultrasound studies with East Asian patients demonstrated that the regression of coronary atherosclerosis could be achieved by moderate-intensity statin therapy [8-10]. Thus, for generalizability of our results to other ethnicities, we added a text in the limitation section as below. As mentioned above, to respond your suggestion, we added two references that you mentioned in the ‘Discussion’ session. Even though those studies provided the clinical relevance of high-intensity statin in elderly population, they also have some limitations. Their study population was not homogenous AMI patients unlike our study. About 60% of study population were CAD not knowing information about revascularization in these studies.[11] Besides, with cautiously looking at the meta-analysis by Cholesterol Treatment Trialists’ Collaboration, 26 trials were analyzed statin therapy versus control, not comparison the intensity of statin.[12] Thus, we need well-powered large scale randomized studies regarding this topic.

Accordingly, although our findings may not be conclusive due to inherent limitations, we think that our results can provide an additional evidence for the statin intensity in elderly patients with AMI. Thanks a lot. 

Original text : Line 298, discussion

Kwak A. et al.[19] reported that moderate-intensity statin can be more protective against ischemic cardiovascular and cerebrovascular events compared with high-intensity statin in patients aged 75 years and older. Foody J.M. et al.[20] showed that statin therapy was associated with a significant reduction in all-cause mortality in patients younger than 80, but not in patients aged 80 and older. Likewise, our findings are in line with the results of these previous studies.

Modified text :

IDEAL (Incremental DEcrease through Aggressive Lipid Lowering) Study randomly assigned 8,888 patients age ≤ 80 years with previous myocardial infarction to receive either intensive therapy with atorvastatin 80 mg/day or the standard statin therapy at the time (simvastatin 20 to 40 mg/day) to compare the ability of these regimens to lower cardiovascular risk. In the subgroup analysis, significant reductions in primary and secondary endpoints were observed only in patients <65 years of age compared to 65 to 80 years of age.[24] Kwak A. et al.[25] reported that moderate-intensity statin can be more protective against ischemic cardiovascular and cerebrovascular events compared with high-intensity statin in patients aged 75 years and older. Foody J.M. et al.[26] showed that statin therapy was associated with a significant reduction in all-cause mortality in patients younger than 80, but not in patients aged 80 and older. Although, some analyses reported an association between high-intensity statin use and a survival benefit in older adults recently,[27, 28] their study population was not homogenous AMI patients unlike our study. About 60% of study population were CAD not knowing information about revascularization in these studies.[27] Besides, with cautiously looking at the meta-analysis by Cholesterol Treatment Trialists’ Collaboration, 26 trials were analyzed statin therapy versus control, not comparison the intensity of statin.[28] Thus, we need well-powered large scale randomized studies regarding this topic.

Original text : Line 334, discussion

Modified text : 

5) the study enrolled only South Korean population. Whether the results of the current study can be extrapolated to other ethnicities is unknown. Caution is needed in extrapolating these results outside of South Korea.

Reference

1. Stone NJ, Robinson J, Lichtenstein AH, Merz CNB, Blum CB, Eckel RH, et al. 2013 ACC/AHA guideline on the treatment of blood cholesterol to reduce atherosclerotic cardiovascular risk in adults: a report of the American College of Cardiology/American Heart Association Task Force on Practice Guidelines. Circulation. 2014;129(25 Suppl 2):S1–45.

2. Grundy SM, Stone NJ, Bailey AL, Beam C, Birtcher KK, Blumenthal RS, et al. 2018 AHA/ACC/AACVPR/AAPA/ABC/ACPM/ADA/AGS/APhA/ASPC/NLA/PCNA Guideline on the Management of Blood Cholesterol: A Report of the American College of Cardiology/American Heart Association Task Force on Clinical Practice Guidelines. Circulation. 2019;139(25):e1082-e143

3. Christopher P. Cannon, M.D., Eugene Braunwald, M.D., Carolyn H. et al. Intensive versus Moderate Lipid Lowering with Statins after Acute Coronary Syndromes. PROVE IT–TIMI 22 study. N Engl J Med 2004;350:1495-504.

4. Matti J Tikkanen 1, Ingar Holme, Nilo B Cater, Michael Szarek, Ole Faergeman et al. Comparison of efficacy and safety of atorvastatin (80 mg) to simvastatin (20 to 40 mg) in patients aged <65 versus >or=65 years with coronary heart disease (from the Incremental DEcrease through Aggressive Lipid Lowering [IDEAL] study). Am J Cardiol 2009;103:577-82

5. Park MW, Park GM, Han S, Yang Y, Kim YG, Roh JH, et al. Moderate-intensity versus high-intensity statin therapy in Korean patients with angina undergoing percutaneous coronary intervention with drug-eluting stents: A propensity-score matching analysis. PloS one. 2018;13(12):e0207889

6. Lee E, Ryan S, Birmingham B, Zalikowski J, March R, Ambrose H, et al. Rosuvastatin pharmacokinetics and pharmacogenetics in white and Asian subjects residing in the same environment. Clin Pharmacol Ther. 2005; 78: 330–341

7. Kong SH, Koo BK, Moon MK. Efficacy of Moderate Intensity Statins in the Treatment of Dyslipidemia in Korean Patients with Type 2 Diabetes Mellitus. Diabetes Metab J. 2017; 41: 23–30. 

8. Lee CW, Kang SJ, Ahn JM, Song HG, Lee JY, Kim WJ, et al. Comparison of effects of atorvastatin (20mg) versus rosuvastatin (10 mg) therapy on mild coronary atherosclerotic plaques (from the ARTMAP trial). Am J Cardiol. 2012; 109: 1700–1704. 

9. Hiro T, Kimura T, Morimoto T, Miyauchi K, Nakagawa Y, Yamagishi M, et al. Effect of intensive statin therapy on regression of coronary atherosclerosis in patients with acute coronary syndrome: a multicenter randomized trial evaluated by volumetric intravascular ultrasound using pitavastatin versus atorvastatin (JAPAN-ACS [Japan assessment of pitavastatin and atorvastatin in acute coronary syndrome] study). J Am Coll Cardiol. 2009; 54: 293–302. 

10. Takayama T, Hiro T, Yamagishi M, Daida H, Hirayama A, Saito S, et al. Effect of rosuvastatin on coronary atheroma in stable coronary artery disease: multicenter coronary atherosclerosis study measuring effects of rosuvastatin using intravascular ultrasound in Japanese subjects (COSMOS). Circ J. 2009;73: 2110–2117. 

11. Rodriguez F, Maron DJ, Knowles JW, Virani SS, Lin S, Heidenreich PA. Association Between Intensity of Statin Therapy and Mortality in Patients With Atherosclerotic Cardiovascular Disease. JAMA cardiology. 2017;2(1):47-54. Epub 2016/11/10. doi: 10.1001/jamacardio.2016.4052. PubMed PMID: 27829091.

12. Efficacy and safety of statin therapy in older people: a meta-analysis of individual participant data from 28 randomised controlled trials. Lancet (London, England). 2019;393(10170):407-15. Epub 2019/02/05. doi: 10.1016/s0140-6736(18)31942-1. PubMed PMID: 30712900; PubMed Central PMCID: PMCPMC6429627.

3. The higher risk of clinical events within the first month in the high-intensity statin treated group points to the likely selection bias inherent in this sort of study design. Further supports the likelihood that cannot ascribe the clinical events observed to a treatment effect of the statin.

Answer : As you pointed out, there was an inherent limitation of selection bias which was inevitable issue of observation study. As descripted in the manuscript, in acute phase of AMI, the ischemic complication (ie, cardiac death, MI or stent thrombosis etc…) frequently occurs in spite of optimal medical treatment including statin, because the status of patients suffering from AMI is still unstable and several factors may affect the early period of clinical outcomes. To minimize bias, we investigated monthly clinical events and performed the Cochran-Armitage trend test was used to determine difference in trends in incidence changes of event rates between acute and maintenance phase. As described in Appendix Figure 1, Cochran-Armitage trend testing showed that incidence rates were high in acute phase (<1 month) and decreased in maintenance phase (from 1 month to 12 months) regarding TVF, CV death and TV-MI. So, we decided to perform 1-month landmark analysis. We could demonstrate that the efficacy of high-intensity statin for secondary prevention in overall and younger patients with 1-month landmark analysis.

Original text : Line 213, results

We also presented the monthly incidence rates of clinical outcomes in overall and high- or low-to-moderate-intensity groups to identify a trend between acute and maintenance phase (Appendix Figure 1). As described in Appendix Figure 1, Cochran-Armitage trend testing revealed these trends to be statistically significant, with p value less than 0.001 in terms of CV death and TV-MI.

Modified text :

We also investigated the monthly incidence rates of clinical outcomes in overall population to identify a discriminating trends between acute and maintenance phase (Appendix Figure 1). As described in Appendix Figure 1, Cochran-Armitage trend testing revealed these trends to be statistically significant in terms of CV death and TV-MI (p<0.001), which showed incidence rates were high in acute phase (<1 month) and decreased in maintenance phase (from 1 month to 12 months).

Modified text : supplemental material

Appendix Figure 1. Monthly incidence rates for clinical outcomes.

Original text : Line 325, discussion

1) This was a non-randomized, observational registry study with inherent methodological limitations; thus, its overall findings must be considered as hypothesis generating only.

Modified text :

1) This was a non-randomized, observational registry study with inherent methodological limitations; thus, its overall findings must be considered as hypothesis generating only. However, even though an inherent limitation, we could demonstrate that the efficacy of high-intensity statin for secondary prevention in younger patients with 1-month landmark analysis. This finding is well in line with the evidence provided by the current guidelines,[12, 13] in which means the clinical relavance of our result.

4. We have seen from PROSPER, PALM and other studies that statins are generally quite well tolerated in older adults. While there are concerns about polypharmacy, frailty and drug interactions, the benefits likely outweigh the risks of high-intensity statin treatment in older adults with known ASCVD.

Answer : Thanks for reviewer's comment. There have been several evidences of tolerance of statin in elderly patients. We presented the proportion of patients who continued the same intensity of statin in both younger and elderly patients. We added this in 'result’ and 'discussion’ session.

Original text : Line 261, results

Modified text :

Subgroup analysis for clinical impact of statin-intensity among patients continuing statin-intensity

In order to further support the results of our study, we performed subgroup analysis with patients who were continuing the same intensity of statin. Because adherence and tolerance are important clinical aspects of medical treatment that affects clinical outcomes,[17] we investigated the maintenance proportion of the same intensity of statin during 1 year after index AMI in total of 6,566 patients. In Figure 2, in patients who received high-intensity statin at discharge, the proportion of patients who maintained the same intensity of statin was significantly lower in elderly patients than in younger patients (55.0% vs. 66.3%, p=0.0009), which was not shown in low-to-moderate-intensity statin group (99.1% vs. 98.1%, p=0.06). In other words, de-escalation of statin-intensity was frequently occurred in elderly patients treated with high-intensity statin.

Original text : Line 271, discussion

2) in acute phase, there were higher risk of adverse clinical events in the high-intensity statin group

Modified text : 

2) in patients who received high-intensity statin at discharge, the proportion of patients who maintained the same intensity of statin was significantly lower in elderly patients than that of younger patients;

Reviewer #2: 1. The change of the amount of statin during follow-up due to side effects or lower the cholesterol level further is very common but not showed in this study. Cholesterol level at follow-up and use of rate of ezetimibe or PCSK9 did not show. Those variables can affect the clinical outcomes.

 Answer : I appreciated with reviewer’s valuable comment. With response to your comment, we investigated the change of statin intensity during 1 year after index AMI with a total of 6,566 patients. In patients who received high-intensity statin at discharge, the proportion of patients who maintained the same intensity of statin during 1 year was significantly lower in elderly patients than in younger patients (55.0% vs. 66.3%, p=0.0009), which was not shown in less-intensive statin group (99.1% vs. 98.1%, p=0.06) (please see below figure). In other words, de-escalation of statin-intensity was frequently observed in elderly patients treated with high-intensity statin.

In overall and younger patients, patients who maintained high-intensity statin showed significantly better clinical outcome in terms of TVF (overall: HR 0.64, 95% CI 0.46-0.9, p=0.01; younger patients: HR 0.63, 95% CI 0.44-0.9, p=0.012). However, in elderly patients, the efficacy for reducing adverse clinical outcomes did not differ between continuous use of high- and low-to-moderate-intensity statin (HR 0.7, 95% CI 0.28-1.77, p=0.46) (please see below table). These findings were consistent with the main result of our study. However, the incidence rate of TVF was numerically lower in high-intensity statin group compared to low-to-moderate-intensity statin group. To overcome the small number of event rates, we extended a study period to 24 months, which showed a consistent result (Appendix Table 5). 

However, unfortunately, there were no available data regarding side effects of statin or prescription of ezetimibe, which was limitation of our study. While, PCSK9 inhibitor was not widely used due to insurance problem in the study period. We did not show cholesterol level at follow-up because of missing data. We added this in the ‘Limitation’ session. Thanks.

Original text : Line 261, results

Modified text :

Subgroup analysis for clinical impact of statin-intensity among patients continuing statin-intensity

In order to further support the results of our study, we performed subgroup analysis with patients who was continuing the same intensity of statin. Because adherence and tolerance are important clinical aspects of medical treatment that affects clinical outcomes,[17] we investigated the maintenance proportion of the same intensity of statin during 1 year with a total of 6,566 patients. In Figure 2, in patients who received high-intensity statin at discharge, the proportion of patients who maintained the same intensity of statin was significantly lower in elderly patients than in younger patients (55.0% vs. 66.3%, p=0.0009), which was not shown in less-intensive statin group (99.1% vs. 98.1%, p=0.06). In other words, de-escalation of statin-intensity was frequently observed in elderly patients treated with high-intensity statin.

Figure 2. Age differences on maintenance proportion of statin-intensity.

Appendix Table 5. Subgroup analysis of event rates and hazard ratios for target-vessel failure among patients with continuing statin-intensity at 1 year.

Original text : Line 331, discussion

4) To verify deterioration of endogenous cholesterol synthesis with increasing age, it is necessary to comfirm the markers of cholesterol metabolism such as lathosterol, campesterol and sitosterol and their ratios to cholesterol. However, this registry did not identify those kinds of markers.

Modified text :

4) No data was available regarding side effects of statin, prescription of ezetimibe. And PCSK9 inhibitor was not widely used in the study period due to governmental insurance coverage.

2. In this study did not showed combined outcomes between 0 to 1-month and 1 to 12-month together in manuscript, which seems to be more important.

Answer : We already provided the result of combined outcomes in Appendix Table 1, which clinical outcomes were not different according to statin-intensity in overall population. So, we additionally analyzed with subgroup who was continuing the same statin-intensity at 1 year, which showed favor result for high-intensity statin in overall and younger patients (overall: HR 0.63, 95% CI 0.45-0.87, p=0.005; younger patients: HR 0.63, 95% CI 0.44-0.89, p=0.009). In elderly patients, the incidence rate of TVF was numerically lower in high-intensity statin group compared to low-to-moderate intensity statin group (HR 0.6, 95% CI 0.24-1.51, p=0.28). After 1-month landmark analysis, it showed consistent result. We provided this result in the manuscript and supplement.

Original text : Line 262, results

Modified text :

In subgroup analysis, patients who maintained high-intensity statin showed significantly better clinical outcome in terms of TVF in overall and younger patients (overall: HR 0.64, 95% CI 0.46-0.9, p=0.01; younger patients: HR 0.63, 95% CI 0.44-0.9, p=0.012). However, in elderly patients, the efficacy for reducing adverse clinical outcomes did not differ between continuous use of high- and low-to-moderate-intensity statin (HR 0.7, 95% CI 0.28-1.77, p=0.46). These findings were consistent with the main result of our study. However, the incidence rate of TVF was numerically lower in high-intensity statin group compared to low-to-moderate-intensity statin group. To overcome the small number of event rates, we extended a study period to 24 months, which showed a consistent result (Appendix Table 5).

Appendix Table 5. Subgroup analysis of event rates and hazard ratios for target-vessel failure among patients with continuing statin-intensity at 1 year.

3. This study compared TVF in high dose statin group of >75 was higher than low to moderate dose, even statistically insignificant. Explanations are needed.

Answer : Thank you for valuable comment. Owing to the relatively limited number, some result of this study may not be sufficiently powered to compare between statin-intensity groups in elderly patients. Especially in elderly patients, high-intensity statin showed numerically higher incidence rates of TVF without statistical significance in maintenance phase. To minimize the bias, we additionally performed subgroup analysis with patients who were continuing the same statin-intensity during 1 year. As already answered in the response to your comment #1, in elderly patients with continuing high-intensity statin, the incidence rate of TVF was numerically lower compared to less-intensive statin group, which was consistent through extended study period of 24 months. Finally, to respond your comment, we added a text the limitation session as below. 

 Original text : Line 330, discussion

Modified text : 

3) Owing to the relatively limited number, some result of this study may not be sufficiently powered to compare between statin-intensity groups. Especially in elderly patients, high-intensity statin showed numerically higher incidence rates of TVF without statistical significance in maintenance phase. To minimize the bias, we additionally performed subgroup analysis with patients who was continuing the same statin-intensity during 1 year. In elderly patients with continuing high-intensity statin, the incidence rate of TVF was numerically lower compared to less-intensive statin group, which was consistent through extended study period of 24 months.

4. The positive results of this study are marginal, which is hard to lead to the conclusion.

Answer : With rigorous PS matching and subgroup analysis, the positive results were consistent, which showed better clinical outcomes of high-intensity statin in overall and younger patients. Especially, with subgroup analysis with continuing statin-intensity group, the statistical significance was strong enough (p=0.01 in overall and p=0.012 in younger patients). Thanks. 

Modified text : descripted in No #2.

5. In discussion, page 14 line 20, 1-month seems to be change into 1-year.

 Answer : We divided in acute phase and maintenance phase to estimate accurate efficacy of statin therapy with minimizing inherent selection bias. So, 1-month seem to be correct.

---

## [Decision Letter · Decision Letter 1]

22 Oct 2021

PONE-D-21-14585R1Clinical Impact of Statin Intensity According to Age in Patients with Acute Myocardial InfarctionPLOS ONE

Dear Dr. Park,

Thank you for submitting your manuscript to PLOS ONE. After careful consideration, we feel that it has merit but does not fully meet PLOS ONE’s publication criteria as it currently stands. Therefore, we invite you to submit a revised version of the manuscript that addresses the points raised during the review process.

ACADEMIC EDITOR: The authors have addressed the queries from the reviewers, however the paper requires a thorough review for proper English language.  Please ensure that your decision is justified on PLOS ONE’s publication criteria and not, for example, on novelty or perceived impact.

We look forward to receiving your revised manuscript.

Kind regards,

R. Jay Widmer

Academic Editor

PLOS ONE

Journal Requirements:

Additional Editor Comments (if provided):

The authors have addressed all comments and questions posed by the reviewers, however the manuscript requires a thorough revision for proper English diction and syntax. The authors should carefully review and edit their manuscript to make it clear and concise for the reader.

Reviewers' comments:

Reviewer's Responses to Questions

**Comments to the Author**

1. If the authors have adequately addressed your comments raised in a previous round of review and you feel that this manuscript is now acceptable for publication, you may indicate that here to bypass the “Comments to the Author” section, enter your conflict of interest statement in the “Confidential to Editor” section, and submit your "Accept" recommendation.

Reviewer #1: All comments have been addressed

Reviewer #2: All comments have been addressed

2. Is the manuscript technically sound, and do the data support the conclusions?

Reviewer #1: Yes

Reviewer #2: Partly

3. Has the statistical analysis been performed appropriately and rigorously? 

Reviewer #1: Yes

Reviewer #2: No

4. Have the authors made all data underlying the findings in their manuscript fully available?

Reviewer #1: Yes

Reviewer #2: Yes

5. Is the manuscript presented in an intelligible fashion and written in standard English?

Reviewer #1: No

Reviewer #2: Yes

6. Review Comments to the Author

Reviewer #1: (No Response)

7. PLOS authors have the option to publish the peer review history of their article (what does this mean?). If published, this will include your full peer review and any attached files.

Reviewer #1: No

Reviewer #2: No

---

## [Author Response · Author response to Decision Letter 1]

22 Nov 2021

We thank to the editor and reviewers for their time and input, and their precious comments.

Journal Requirements:

 Answer : Thank you for your encouraging comments and generous understanding of our study. We have tried our best to address the issues you pointed out. Thank you. We conducted an in-depth review of the references and strictly checked. Fortunately, there was no retracted article but, typo errors, which were all corrected.

 Modified text :

References

9. Oliveira JS, Pinheiro MB, Fairhall N, Walsh S, Chesterfield Franks T, Kwok W, et al. Evidence on Physical Activity and the Prevention of Frailty and Sarcopenia Among Older People: A Systematic Review to Inform the World Health Organization Physical Activity Guidelines. Journal of physical activity & health. 2020;17(12):1247-1258. Epub 2020/08/12. doi: 10.1123/jpah.2020-0323. PubMed PMID: 32781432.

20. Nanna MG, Navar AM, Wang TY, Mi X, Virani SS, Louie MJ, et al. Statin Use and Adverse Effects Among Adults >75 Years of Age: Insights From the Patient and Provider Assessment of Lipid Management (PALM) Registry. Journal of the American Heart Association. 2018;7(10):e008546. Epub 2018/05/10. doi: 10.1161/jaha.118.008546. PubMed PMID: 29739801; PubMed Central PMCID: PMC6015311.

26. Cholesterol Treatment Trialists' Collaboration. Efficacy and safety of statin therapy in older people: a meta-analysis of individual participant data from 28 randomised controlled trials. Lancet (London, England). 2019;393(10170):407-15. Epub 2019/02/05. doi: 10.1016/s0140-6736(18)31942-1. PubMed PMID: 30712900; PubMed Central PMCID: PMC6429627.

28. Bach RG, Cannon CP, Giugliano RP, White JA, Lokhnygina Y, Bohula EA, et al. Effect of Simvastatin-Ezetimibe Compared With Simvastatin Monotherapy After Acute Coronary Syndrome Among Patients 75 Years or Older: A Secondary Analysis of a Randomized Clinical Trial. JAMA cardiology. 2019;4(9):846-54. Epub 2019/07/18. doi: 10.1001/jamacardio.2019.2306. PubMed PMID: 31314050; PubMed Central PMCID: PMC6647004.

Additional Editor Comments (if provided):

The authors have addressed all comments and questions posed by the reviewers, however the manuscript requires a thorough revision for proper English diction and syntax. The authors should carefully review and edit their manuscript to make it clear and concise for the reader.

Answer : We performed a thorough revision for proper English diction and syntax. To prove this, we submitted the ‘Editing Certificate’ of American Journal Experts. The modified parts of text are highlighted in blue so that you can recognize easily.

Modified text : 

Line 64

Meanwhile, the efficacy to prevent adverse clinical events between high- and low-to-moderate-intensity statin was not statistically different in elderly patients. Further randomized study with large elderly population is warranted.

Line 119

We compared the efficacy on clinical outcomes according to statin-intensity in younger (<75 years old) and elderly (≥75 years old) patients, respectively.

 Line 210

Adverse clinical outcomes were similar between two statin-intensity groups except for TLR,

 Line 216

As described in Appendix Figure 1, Cochran-Armitage trend testing revealed these trends to be statistically significant in terms of CV death and TV-MI (p<0.001), which showed high incidence rates in acute phase (<1 month) and decreased trend in maintenance phase (from 1 month to 12 months).

 Line 270

In high-intensity statin group, the proportion of patients continuing the same statin-intensity was significantly lower in elderly patients than that of younger patients (55.0% vs. 66.3%, p=0.0009), while this phenomenon was not observed in low-to-moderate-intensity statin group (99.1% vs. 98.1%, p=0.06) (Figure 2).

 Line 294

2) in high-intensity statin group, the proportion of patients continuing the same statin-intensity was significantly lower in elderly patients than that of younger patients;

 Line 332

IDEAL (Incremental DEcrease through Aggressive Lipid Lowering) Study randomly assigned 8,888 patients age ≤ 80 years with previous myocardial infarction to receive either intensive therapy with atorvastatin 80 mg/day or the standard statin therapy (simvastatin 20 to 40 mg/day) to compare the ability of these regimens to lower cardiovascular risk.

 Line 342

Although some analyses reported an association between high-intensity statin use and a survival benefit in older adults,[25, 26] the study population was heterogenous while our study population consisted of only AMI patients. In one retrospective cohort analysis, even about 60% of CAD patients were included in the study population, there is no information of severity or revascularization of coronary lesions.[25] Besides, in the meta-analysis by Cholesterol Treatment Trialists’ Collaboration, 26 trials were based on analysis between statin therapy versus control, not comparison of statin-intensity.[26]

 Line 368

However, even though there is an inherent limitation, we could demonstrate the efficacy of high-intensity statin for secondary prevention in younger patients. This finding is well in line with the evidence provided by the current guidelines.[12, 13]

 Line 385

4) No data was available regarding side effects of statin and prescription rates of ezetimibe or PCSK9 inhibitor. Especially, PCSK9 inhibitor was not widely used in the study period due to governmental insurance coverage.

 Line 396

Meanwhile, the efficacy to prevent adverse clinical events between high- and low-to-moderate-intensity statin was not statistically different in elderly patients. Further randomized study with large elderly population is warranted.

---

## [Decision Letter · Decision Letter 2]

23 Dec 2021

PONE-D-21-14585R2Clinical Impact of Statin Intensity According to Age in Patients with Acute Myocardial InfarctionPLOS ONE

Dear Dr. Park,

Thank you for submitting your manuscript to PLOS ONE. After careful consideration, we feel that it has merit but does not fully meet PLOS ONE’s publication criteria as it currently stands. Therefore, we invite you to submit a revised version of the manuscript that addresses the points raised during the review process.

The authors should carefully respond to each comment raised by the reviewer in an individual fashion in the response/cover letter. 

We look forward to receiving your revised manuscript.

Kind regards,

R. Jay Widmer

Academic Editor

PLOS ONE

Additional Editor Comments (if provided):

Multiple reviewers have read and commented on this paper. There remain some concerns regarding the results and interpretation. Please review the comments and attempt to address each one individually.

Reviewers' comments:

Reviewer's Responses to Questions

**Comments to the Author**

1. If the authors have adequately addressed your comments raised in a previous round of review and you feel that this manuscript is now acceptable for publication, you may indicate that here to bypass the “Comments to the Author” section, enter your conflict of interest statement in the “Confidential to Editor” section, and submit your "Accept" recommendation.

Reviewer #3: (No Response)

2. Is the manuscript technically sound, and do the data support the conclusions?

Reviewer #3: Partly

3. Has the statistical analysis been performed appropriately and rigorously? 

Reviewer #3: No

4. Have the authors made all data underlying the findings in their manuscript fully available?

Reviewer #3: Yes

5. Is the manuscript presented in an intelligible fashion and written in standard English?

Reviewer #3: Yes

6. Review Comments to the Author

Reviewer #3: Lee et all used the COREA database to examine the association between statin use and outcomes following AMI in older adults. I have several major concerns that limit the interpretation of this study.

The authors review of the literature is that there is limited data to guide treatment decisions post-AMI in older adults, however this should be tempered and the rationale for this study should be clarified. The cited guidelines have the greatest ambiguity in relation to primary prevention, not statin use after AMI where time to benefit is short – on the order of days/months, not years as in primary prevention. The recommendation for lower-intensity statins in older adults is not strong and is based on only perceived risk of adverse effects (Mach EHJ 2020).

More importantly, the choice of methods are unclear and make the interpretation of the findings challenging – what is the rationale for running crude models or simple multivariate models? It is well established in pharmacoepidemiology, specifically in statin research, that these methods are insufficient – those who are prescribed a statin are likely at higher risk of an event, exactly as is shown in this study. Confounding by indication must be accounted for and propensity score methods should be the primary analysis. See for example: Seeger JD, Kurth T, Walker AM. Use of propensity score technique to account for exposure-related covariates: an example and lesson. Med Care. 2007;45(10)(suppl 2):S143-S148. doi:10.1097/MLR.0b013e318074ce79

Additionally, the choice of using matching for the PS should be justified. This excludes all patients who could not be matched and may bias the results. See Elze MC, Gregson J, Baber U, et al. Comparison of propensity score methods and covariate adjustment: evaluation in 4 cardiovascular studies. J Am Coll Cardiol. 2017;69(3):345-357. doi:10.1016/j.jacc.2016.10.060 for alternatives and consider IPTW or Overlap Weighting methods that make use of the whole cohort.

The primary outcome needs to be clarified – is it a composite of TVF, CV death, TV-MI, TLR, or are these co-primary? If they are co-primary, there needs to be correction for multiple testing. If it is a composite, the main results should be interpreted as the composite and only secondarily should individual findings be highlighted. Similarly, findings should be tempered accordingly.

Minor: Avoid the use of trial language – this is an observational study and associations are being studied, not effects or efficacy.

7. PLOS authors have the option to publish the peer review history of their article (what does this mean?). If published, this will include your full peer review and any attached files.

Reviewer #3: No

---

## [Author Response · Author response to Decision Letter 2]

6 Feb 2022

Revision Response Letter

Thank you for your encouraging comments and generous understanding of our study. We have tried our best to address the issues you pointed out. Thank you.

REVIEWER #3 :

1. The authors review of the literature is that there is limited data to guide treatment decisions post-AMI in older adults, however this should be tempered and the rationale for this study should be clarified. The cited guidelines have the greatest ambiguity in relation to primary prevention, not statin use after AMI where time to benefit is short – on the order of days/months, not years as in primary prevention. The recommendation for lower-intensity statins in older adults is not strong and is based on only perceived risk of adverse effects (Mach EHJ 2020).

Answer : We appreciate your valuable comment. Through the in-depth review of references, we modified original references into more suitable references. We also tried our best to temper and clarify the rationale of our study as follows;

 Original text : Background, Line 76

Thus, current guidelines recommend that high-intensity statin should be considered based on lowering low-density lipoprotein cholesterol (LDL-C) for secondary prevention in patients with AMI.

 Especially, because of globally increasing number of adults aged 75 years or older and inherent their high cardiovascular risk and frailty,

Modified text :

Thus, current guidelines recommend that high-intensity statin should be considered based on lowering low-density lipoprotein cholesterol (LDL-C) for secondary prevention in patients with atherosclerotic cardiovascular disease (ASCVD).

 Because of globally increasing number of adults aged 75 years or older and inherent their high cardiovascular risk and frailty,

Original text : Background, Line 99

Therefore, to address this issue, we sought to compare the clinical efficacy of high-intensity versus low-to-moderate-intensity statin according to age (<75 or ≥75 years old) using large prospective AMI registry.

Modified text : 

Therefore, we sought to (1) investigate the prescription rate of high-intensity statin and (2) compare the association between statin-intensity (high-intensity versus low-to-moderate-intensity statin) and clinical outcomes according to age (<75 or ≥75 years old) using large prospective AMI registry.

 Original text : References, Line 410

 1. Ridker PM. The JUPITER trial: results, controversies, and implications for prevention. Circulation Cardiovascular quality and outcomes. 2009;2(3):279-85. Epub 2009/12/25. doi: 10.1161/circoutcomes.109.868299. PubMed PMID: 20031849.

 Modified text :

 1. Collet JP, Thiele H, Barbato E, Barthélémy O, Bauersachs J, Bhatt DL, et al. 2020 ESC Guidelines for the management of acute coronary syndromes in patients presenting without persistent ST-segment elevation. European heart journal. 2021;42(14):1289-367. Epub 2020/08/30. doi: 10.1093/eurheartj/ehaa575. PubMed PMID: 32860058.

 Original text : References, Line 413

 2. Baigent C, Blackwell L, Emberson J, Holland LE, Reith C, Bhala N, et al. Efficacy and safety of more intensive lowering of LDL cholesterol: a meta-analysis of data from 170,000 participants in 26 randomised trials. Lancet (London, England). 2010;376(9753):1670-81. Epub 2010/11/12. doi: 10.1016/s0140-6736(10)61350-5. PubMed PMID: 21067804; PubMed Central PMCID: PMC2988224.

3. Fulcher J, O'Connell R, Voysey M, Emberson J, Blackwell L, Mihaylova B, et al. Efficacy and safety of LDL-lowering therapy among men and women: meta-analysis of individual data from 174,000 participants in 27 randomised trials. Lancet (London, England). 2015;385(9976):1397-405. Epub 2015/01/13. doi: 10.1016/s0140-6736(14)61368-4. PubMed PMID: 25579834.

4. Mills EJ, Rachlis B, Wu P, Devereaux PJ, Arora P, Perri D. Primary prevention of cardiovascular mortality and events with statin treatments: a network meta-analysis involving more than 65,000 patients. Journal of the American College of Cardiology. 2008;52(22):1769-81. Epub 2008/11/22. doi: 10.1016/j.jacc.2008.08.039. PubMed PMID: 19022156.

 Modified text :

2. Amarenco P, Bogousslavsky J, Callahan A, 3rd, Goldstein LB, Hennerici M, Rudolph AE, et al. High-dose atorvastatin after stroke or transient ischemic attack. The New England journal of medicine. 2006;355(6):549-59. Epub 2006/08/11. doi: 10.1056/NEJMoa061894. PubMed PMID: 16899775.

3. Baigent C, Blackwell L, Emberson J, Holland LE, Reith C, Bhala N, et al. Efficacy and safety of more intensive lowering of LDL cholesterol: a meta-analysis of data from 170,000 participants in 26 randomised trials. Lancet (London, England). 2010;376(9753):1670-81. Epub 2010/11/12. doi: 10.1016/s0140-6736(10)61350-5. PubMed PMID: 21067804; PubMed Central PMCID: PMCPMC2988224.

2. More importantly, the choice of methods are unclear and make the interpretation of the findings challenging – what is the rationale for running crude models or simple multivariate models? It is well established in pharmacoepidemiology, specifically in statin research, that these methods are insufficient – those who are prescribed a statin are likely at higher risk of an event, exactly as is shown in this study. Confounding by indication must be accounted for and propensity score methods should be the primary analysis. See for example: Seeger JD, Kurth T, Walker AM. Use of propensity score technique to account for exposure-related covariates: an example and lesson. Med Care. 2007;45(10)(suppl 2):S143-S148. doi:10.1097/MLR.0b013e318074ce79

Additionally, the choice of using matching for the PS should be justified. This excludes all patients who could not be matched and may bias the results. See Elze MC, Gregson J, Baber U, et al. Comparison of propensity score methods and covariate adjustment: evaluation in 4 cardiovascular studies. J Am Coll Cardiol. 2017;69(3):345-357. doi:10.1016/j.jacc.2016.10.060 for alternatives and consider IPTW or Overlap Weighting methods that make use of the whole cohort.

Answer : We appreciate your valuable comment. As you recommended, we performed IPTW analysis for the primary analysis in order to minimize the bias. And, we added the results in a main text and changed tables and figures as follows. Thanks again.

Original text : Methods, Line 172

For all crude, multivariable-adjusted analyses, treatment effects were evaluated in all the patients and in each group of younger and elderly patients. 

 Modified text :

Inverse probability of treatment weighting (IPTW) based on the propensity score (probability of receiving high-intensity statin) was used as the primary tool to adjust for differences in the baseline characteristics between the high-intensity and low-to-moderate intenstiy groups. Once each patient’s propensity score was estimated, weights were calculated using the method described in the previous literature.[16] We examined the similarity of the baseline characteristics between the treatment groups before and after IPTW.[17] After confirming the comparability of the two groups in the data with IPTW, we ran the Cox proportional hazard model and made statistical inference using robust standard errors (Huber sandwich estimator).[18]

For all crude, multivariable-adjusted, and IPTW analyses, treatment effects were evaluated in overall patients and in each group of younger and elderly patients.

 Original text : Results, Line 205

Involvement of proximal left anterior descending artery was shown frequently in high-intensity statin group among younger patients (Table 1).

Modified text :

Involvement of proximal left anterior descending artery was shown frequently in high-intensity statin group among younger patients (Table 1). As shown in Table 2 and Appendix Table 2, majority of those differences of covariates were balanced after PS matching and IPTW.

 Original text : Tables 

 Modified text : 

Table 2. Demographics after inverse probability weighting.

 <75 years old ≥75 years old

Characteristics Less intensive

(n = 6133) High-intensity

(n = 5683) P SMD Less-intensity

(n = 1548) High-intensity

(n = 1435) P SMD

Baseline patients characteristics

Age (years) 58.5 ± 10.2 58.3 ± 9.8 0.64 0.015 80.1 ± 5.4 80.0 ± 5.1 0.76 0.019

Male 4878 (79.5) 4592 (80.8) 0.33 0.032 733 (47.4) 691 (48.1) 0.81 0.016

Hypertension 2896 (47.2) 2689 (47.3) 0.96 0.002 1012 (65.4) 915 (63.8) 0.61 0.034

Diabetes mellitus 1841 (30.0) 1687 (29.7) 0.82 0.007 505 (32.6) 470 (32.8) 0.97 0.003

Hyperlipidemia 1068 (17.4) 1137 (20.0) 0.039 0.066 192 (12.4) 212 (14.8) 0.32 0.068

Current smoker 2994 (48.8) 2789 (49.1) 0.88 0.005 235 (15.2) 207 (14.4) 0.74 0.021

Clinical diagnosis 0.55 0.019 0.99 0.002

STEMI 3417 (55.7) 3111 (54.7) 712 (46.0) 659 (45.9) 

NSTEMI 2717 (44.3) 2573 (45.3) 836 (54.0) 777 (54.1) 

CKD 99.3 (1.6) 87.5 (1.5) 0.84 0.006 19 (1.2) 18 (1.3) 0.96 0.003

Prior MI 156 (2.5) 135 (2.4) 0.77 0.011 72 (4.6) 62 (4.4) 0.84 0.014

Prior PCI 290 (4.7) 266 (4.7) 0.94 0.002 127 (8.2) 116 (8.1) 0.95 0.004

Prior CABG 19 (0.3) 10 (0.2) 0.58 0.027 11 (0.7) 11 (0.8) 0.90 0.009

Prior CVA 350 (5.7) 323 (5.7) 0.97 0.001 158 (10.2) 142 (9.9) 0.86 0.011

PAD 24 (0.4) 24 (0.4) 0.90 0.005 12 (0.8) 10 (0.7) 0.92 0.009

AF 121 (2.0) 93 (1.6) 0.53 0.025 70 (4.5) 54 (3.8) 0.65 0.039

LV EF, % 54.2 ± 10.5 54.2 ± 10.7 0.83 0.007 51.0 ± 11.9 50.7 ± 11.6 0.67 0.028

Lesion and Procedural characteristics

Radial access 1244 (20.3) 1212 (21.3) 0.40 0.026 343 (22.1) 333 (23.2) 0.71 0.027

LM involved 353 (5.8) 336 (5.9) 0.84 0.006 139 (9.0) 118 (8.2) 0.70 0.028

pLAD involved 2541 (41.4) 2500 (44.0) 0.11 0.052 707 (45.7) 693 (48.3) 0.44 0.051

Disease extent 0.44 0.052 0.35 0.12

1VD 2939 (47.9) 2715 (47.8) 572 (37.0) 607 (42.3) 

 2VD 1966 (32.1) 1870 (32.9) 539 (34.8) 463 (32.2) 

 3VD 1228 (20.0) 1099 (19.3) 437 (28.2) 366 (25.5) 

Complex PCI 2636 (43.0) 2484 (43.7) 0.65 0.015 705 (45.6) 665 (46.3) 0.82 0.015

Total stent number 1.62 ± 0.88 1.64 ± 0.91 0.47 0.025 1.67 ± 0.85 1.66 ± 0.90 0.87 0.012

Mean stent diameter 3.20 ± 0.42 3.20 ± 0.51 0.97 0.001 3.07 ± 0.35 3.07 ± 0.77 0.94 0.006

Total stent length 34.47 ± 20.97 35.20 ± 22.73 0.34 0.033 35.65 ± 20.66 34.97 ± 21.53 0.65 0.032

IVUS use 1382 (22.5) 1279 (22.5) 0.98 0.001 261 (16.8) 234 (16.3) 0.82 0.015

SMD, standardised mean difference; STEMI, ST-segment elevation myocardial infarction; NSTEMI, non ST-segment elevation myocardial infarction; CKD, chronic kidney disease; MI, myocardial infarction; PCI, percutaneous coronary intervention; CABG, coronary artery bypass grafting surgery; CVA, cerebrovascular attack; PAD, peripheral artery disease; AF, atrial fibrillation; LV EF, left ventricle ejection fraction; HDL, high-density lipoprotein; LDL, low-density lipoprotein; LM, left main; pLAD, proximal left anterior descending artery; VD, vessel disease; IVUS, intravascular ultrasound.

 Original text : Table 2, 3

Table 2. Event rates and hazard ratios for clinical outcomes in acute phase (<1 month).

Outcomes Event Rates at 1 Month (n/%*) Crude Multivariate Adjusted† 

 Low-to-moderate-intensity High-intensity HR (95% CI) P HR (95% CI) P PInteraction

Overall 

TVF 52 (0.8)** 25 (1.3)** 1.6 (1.0-2.58) 0.052 1.72 (1.07-2.78) 0.026 0.76

All-cause death 28 (0.5)** 17 (0.9)** 2.03 (1.11-3.7) 0.022 2.21 (1.21-4.04) 0.01 0.91

CV death 27 (0.4)** 16 (0.9)** 1.98 (1.07-3.67) 0.031 2.16 (1.16-4.02) 0.015 0.65

TV-MI 10 (0.2) 6 (0.3) 2.0 (0.73-5.49) 0.18 2.07 (0.75-5.71) 0.16 0.52

TLR 23 (0.4) 5 (0.3) 0.72 (0.27-1.9) 0.51 0.76 (0.29-2.0) 0.58 0.46

< 75 years old 

TVF 31 (0.6) 16 (1.1) 1.68 (0.92-3.07) 0.09 1.84 (1.00-3.37) 0.048 

All-cause death 16 (0.3) 10 (0.7) 2.04 (0.92-4.49) 0.08 2.34 (1.06-5.17) 0.036 

CV death 15 (0.3) 10 (0.7) 2.17 (0.98-4.84) 0.057 2.49 (1.12-5.56) 0.026 

TV-MI 8 (0.2) 4 (0.3) 1.63 (0.49-5.4) 0.43 1.72 (0.52-5.37) 0.38 

TLR 14 (0.3) 4 (0.3) 0.93 (0.31-2.82) 0.90 0.99 (0.32-3.01) 0.98 

≥ 75 years old 

TVF 21 (1.7) 9 (2.6) 1.57 (0.72-3.44) 0.26 1.57 (0.72-3.42) 0.26 

All-cause death 12 (0.9) 7 (2.0) 2.14 (0.84-5.44) 0.11 2.13 (0.84-5.4) 0.11 

CV death 12 (0.9) 6 (1.7) 1.84 (0.69-4.89) 0.22 1.82 (0.68-4.86) 0.23 

TV-MI 2 (0.2) 2 (0.6) 3.65 (0.51-25.9) 0.20 3.63 (0.51-25.8) 0.20 

TLR 9 (0.7) 1 (0.3) 0.41 (0.05-3.2) 0.39 0.41 (0.05-3.25) 0.40 

†adjusted by covariates including age, diabetes mellitus.

*Event rates were derived from the Kaplan-Meier estimates. Hazard ratio is the risk of high-intensity statin for clinical outcomes compared with that of less intensive statin.

**P value by log-rank test was less than 0.05.

HR, hazard ratio; TVF, target vessel failure; CV, cardiovascular; TV-MI, target-vessel myocardial infarction; TLR, target lesion revascularization; ST, stent thrombosis.

Table 3. Event rates and hazard ratios for clinical outcomes in maintenance phase (from 1 month to 12 months).

Outcomes Event Rates at 1-12 Month (n/%*) Crude Multivariate Adjusted† 

 Low-to-moderate-intensity High-intensity HR (95% CI) P HR (95% CI) P Pinteraction

Overall 

TVF 460 (7.7) 117 (6.6) 0.86 (0.7-1.05) 0.14 0.85 (0.69-1.05) 0.12 0.10

All-cause death 220 (3.7) 71 (4.0) 1.1 (0.84-1.43) 0.50 1.15 (0.87-1.52) 0.33 0.20

CV death 164 (2.7) 53 (3.0) 1.1 (0.8-1.49) 0.56 1.1 (0.79-1.53) 0.58 0.15

TV-MI 38 (0.6) 10 (0.6) 0.89 (0.45-1.79) 0.75 0.83 (0.4-1.73) 0.62 >0.99

TLR 295 (5.0)** 64 (3.7)** 0.73 (0.56-0.96) 0.024 0.73 (0.55-0.96) 0.022 0.78

< 75 years old 

TVF 329 (6.9) 79 (5.5) 0.79 (0.62-1.01) 0.059 0.76 (0.59-0.99) 0.038 

All-cause death 107 (2.2) 32 (2.2) 0.99 (0.67-1.47) 0.95 0.97 (0.63-1.49) 0.89 

CV death 76 (1.6) 22 (1.5) 0.96 (0.59-1.54) 0.85 0.85 (0.5-1.46) 0.56 

TV-MI 27 (0.6) 10 (0.7) 1.22 (0.59-2.53) 0.59 1.11 (0.52-2.39) 0.79 

TLR 250 (5.3)** 55 (3.9)** 0.72 (0.54-0.97) 0.029 0.71 (0.53-0.96) 0.025 

≥ 75 years old 

TVF 131 (10.8) 38 (11.7) 1.1 (0.76-1.58) 0.61 1.1 (0.76-1.59) 0.63 

All-cause death 113 (9.2) 39 (11.8) 1.32 (0.91-1.89) 0.14 1.35 (0.92-1.97) 0.12 

CV death 88 (7.2) 31 (9.5) 1.34 (0.89-2.02) 0.16 1.34 (0.88-2.05) 0.18 

TV-MI 11 (0.9) 0 NA >0.99 NA >0.99 

TLR 45 (3.9) 9 (2.9) 0.76 (0.37-1.55) 0.45 0.76 (0.37-1.57) 0.46 

†adjusted by covariates including diabetes mellitus, chronic renal disease, peripheral artery disease, atrial fibrillation, left ventricle ejection fraction, left main disease, and total stent number.

*Event rates were derived from the Kaplan-Meier estimates. Hazard ratio is the risk of high-intensity statin for clinical outcomes compared with that of less intensive statin.

**P value by log-rank test was less than 0.05.

HR, hazard ratio; TVF, target vessel failure; CV, cardiovascular; TV-MI, target-vessel myocardial infarction; TLR, target lesion revascularization.

 Modified text :

Table 3. Event rates and hazard ratios for clinical outcomes in acute phase (<1 month).

Outcomes Event Rates at 1 Month (n/%*) Crude Multivariate Adjusted† IPTW Adjusted

 Low-to-moderate-intensity High-intensity HR (95% CI) P HR (95% CI) P HR (95% CI) P PInteraction

Overall 

TVF 52 (0.8)** 25 (1.3)** 1.6 (1.0-2.58) 0.052 1.72 (1.07-2.78) 0.026 1.24 (0.73-2.10) 0.43 0.97

All-cause death 28 (0.5)** 17 (0.9)** 2.03 (1.11-3.7) 0.022 2.21 (1.21-4.04) 0.01 1.37 (0.70-2.70) 0.36 0.96

CV death 27 (0.4)** 16 (0.9)** 1.98 (1.07-3.67) 0.031 2.16 (1.16-4.02) 0.015 1.32 (0.65-2.66) 0.44 0.68

TV-MI 10 (0.2) 6 (0.3) 2.0 (0.73-5.49) 0.18 2.07 (0.75-5.71) 0.16 1.72 (0.61-4.86) 0.31 0.31

TLR 23 (0.4) 5 (0.3) 0.72 (0.27-1.9) 0.51 0.76 (0.29-2.0) 0.58 0.65 (0.24-1.78) 0.40 0.73

< 75 years old 

TVF 31 (0.6) 16 (1.1) 1.68 (0.92-3.07) 0.09 1.84 (1.00-3.37) 0.048 1.23 (0.63-2.38) 0.54 

All-cause death 16 (0.3) 10 (0.7) 2.04 (0.92-4.49) 0.08 2.34 (1.06-5.17) 0.036 1.39 (0.57-3.38) 0.47 

CV death 15 (0.3) 10 (0.7) 2.17 (0.98-4.84) 0.057 2.49 (1.12-5.56) 0.026 1.48 (0.60-3.64) 0.39 

TV-MI 8 (0.2) 4 (0.3) 1.63 (0.49-5.4) 0.43 1.72 (0.52-5.37) 0.38 1.16 (0.35-3.90) 0.81 

TLR 14 (0.3) 4 (0.3) 0.93 (0.31-2.82) 0.90 0.99 (0.32-3.01) 0.98 0.73 (0.23-2.29) 0.59 

≥ 75 years old 

TVF 21 (1.7) 9 (2.6) 1.57 (0.72-3.44) 0.26 1.57 (0.72-3.42) 0.26 1.21 (0.51-2.87) 0.66 

All-cause death 12 (0.9) 7 (2.0) 2.14 (0.84-5.44) 0.11 2.13 (0.84-5.4) 0.11 1.37 (0.46-4.04) 0.57 

CV death 12 (0.9) 6 (1.7) 1.84 (0.69-4.89) 0.22 1.82 (0.68-4.86) 0.23 1.08 (0.34-3.37) 0.90 

TV-MI 2 (0.2) 2 (0.6) 3.65 (0.51-25.9) 0.20 3.63 (0.51-25.8) 0.20 3.92 (0.5-30.53) 0.19 

TLR 9 (0.7) 1 (0.3) 0.41 (0.05-3.2) 0.39 0.41 (0.05-3.25) 0.40 0.48 (0.07-3.61) 0.48 

†adjusted by covariates including age, diabetes mellitus.

*Event rates were derived from the Kaplan-Meier estimates. Hazard ratio is the risk of high-intensity statin for clinical outcomes compared with that of less intensive statin.

**P value by log-rank test was less than 0.05.

HR, hazard ratio; TVF, target vessel failure; CV, cardiovascular; TV-MI, target-vessel myocardial infarction; TLR, target lesion revascularization.

Table 4. Event rates and hazard ratios for clinical outcomes in maintenance phase (from 1 month to 12 months).

Outcomes Event Rates at 1-12 Month (n/%*) Crude Multivariate Adjusted† IPTW Adjusted

 Low-to-moderate-intensity High-intensity HR (95% CI) P HR (95% CI) P HR (95% CI) P PInteraction

Overall 

TVF 460 (7.7) 117 (6.6) 0.86 (0.7-1.05) 0.14 0.85 (0.69-1.05) 0.12 0.85 (0.68-1.06) 0.15 0.17

All-cause death 220 (3.7) 71 (4.0) 1.1 (0.84-1.43) 0.50 1.15 (0.87-1.52) 0.33 1.17 (0.86-1.57) 0.32 0.30

CV death 164 (2.7) 53 (3.0) 1.1 (0.8-1.49) 0.56 1.1 (0.79-1.53) 0.58 1.06 (0.75-1.49) 0.76 0.28

TV-MI 38 (0.6) 10 (0.6) 0.89 (0.45-1.79) 0.75 0.83 (0.4-1.73) 0.62 0.84 (0.37-1.92) 0.69 NA

TLR 295 (5.0)** 64 (3.7)** 0.73 (0.56-0.96) 0.024 0.73 (0.55-0.96) 0.022 0.72 (0.54-0.97) 0.033 0.76

< 75 years old 

TVF 329 (6.9) 79 (5.5) 0.79 (0.62-1.01) 0.059 0.76 (0.59-0.99) 0.038 0.75 (0.57-0.99) 0.044 

All-cause death 107 (2.2) 32 (2.2) 0.99 (0.67-1.47) 0.95 0.97 (0.63-1.49) 0.89 0.96 (0.61-1.52) 0.86 

CV death 76 (1.6) 22 (1.5) 0.96 (0.59-1.54) 0.85 0.85 (0.5-1.46) 0.56 0.82 (0.47-1.43) 0.48 

TV-MI 27 (0.6) 10 (0.7) 1.22 (0.59-2.53) 0.59 1.11 (0.52-2.39) 0.79 1.19 (0.51-2.76) 0.69 

TLR 250 (5.3)** 55 (3.9)** 0.72 (0.54-0.97) 0.029 0.71 (0.53-0.96) 0.025 0.70 (0.51-0.97) 0.033 

≥ 75 years old 

TVF 131 (10.8) 38 (11.7) 1.1 (0.76-1.58) 0.61 1.1 (0.76-1.59) 0.63 1.06 (0.72-1.57) 0.76 

All-cause death 113 (9.2) 39 (11.8) 1.32 (0.91-1.89) 0.14 1.35 (0.92-1.97) 0.12 1.33 (0.90-1.97) 0.15 

CV death 88 (7.2) 31 (9.5) 1.34 (0.89-2.02) 0.16 1.34 (0.88-2.05) 0.18 1.24 (0.80-1.92) 0.35 

TV-MI 11 (0.9) 0 NA NA NA 

TLR 45 (3.9) 9 (2.9) 0.76 (0.37-1.55) 0.45 0.76 (0.37-1.57) 0.46 0.81 (0.37-1.75) 0.59 

†adjusted by covariates including age, diabetes mellitus.

*Event rates were derived from the Kaplan-Meier estimates. Hazard ratio is the risk of high-intensity statin for clinical outcomes compared with that of less intensive statin.

**P value by log-rank test was less than 0.05.

HR, hazard ratio; TVF, target vessel failure; CV, cardiovascular; TV-MI, target-vessel myocardial infarction; TLR, target lesion revascularization.

 Original text : Results, Line 222

The incidence rates of clinical outcomes between two statin-intensity groups in the overall population, younger and elderly patients within 1 month after index PCI are shown in Table 2. Kaplan-Meier curves for TVF and TLR at 1 month were shown in Figure 3. In overall patients, early clinical outcomes regarding TVF, all-cause death and CV death were worse in the high-intensity statin group (TVF : aHR, 1.72; 95% CI, 1.07-2.78; p=0.026; all-cause death : aHR, 2.21; 95% CI, 1.21-4.04; p=0.01; CV death : aHR, 2.16; 95% CI, 1.16-4.02; p=0.015). After subgroup analysis by aged of 75, these findings were consistent in younger patients. However, in elderly patients, although there were numerically higher event rates in the high-intensity statin group without statistical significance. No significant interactions were found between aged group and statin intensity in any of the adjusted 1-month risks of study outcomes (Pinteraction = 0.76 for TVF, Pinteraction = 0.65 for CV death, and Pinteraction = 0.46 for TLR).

 Modified text :

The incidence rates of clinical outcomes between two statin-intensity groups in the overall population, younger and elderly patients within 1 month after index PCI are shown in Table 3. Kaplan-Meier curves for TVF at 1 month were shown in Figure 3, adjusted by IPTW analysis. In overall patients, the high-intensity statin group showed poorer 1-month clinical outcomes regarding TVF, all-cause death and CV death (TVF : aHR, 1.72; 95% CI, 1.07-2.78; p=0.026; all-cause death : aHR, 2.21; 95% CI, 1.21-4.04; p=0.01; CV death : aHR, 2.16; 95% CI, 1.16-4.02; p=0.015). However, after IPTW analysis, no differences were observed with statin-intensity in both younger and elderly patients (Table 3). 

After subgroup analysis by aged of 75, these findings were consistent in younger patients. Meanwhile, in elderly patients, despite numerically higher event rates in the high-intensity statin group, there were no statistical significance (Table 3). No significant interactions were found between aged group and statin intensity in any of the adjusted 1-month risks of study outcomes (Pinteraction = 0.97 for TVF, Pinteraction = 0.96 for all-cause death, Pinteraction = 0.68 for CV death, Pinteraction = 0.31 for TV-MI and Pinteraction = 0.73 for TLR).

Original text : Results, Line 236

In overall population, the risk of adverse clinical outcomes according to statin-intensity was not different except for TLR (TVF : aHR, 0.85; 95% CI, 0.69-1.05; p=0.12; all-cause death : aHR, 1.15; 95% CI, 0.87-1.52; p=0.33; TLR : aHR, 0.73; 95% CI, 0.55-0.96; p=0.022). In younger patients, the high-intensity statin group showed significantly better clinical outcomes in terms of TVF (aHR, 0.76; 95% CI, 0.59-0.99; p=0.038) and TLR (aHR, 0.72; 95% CI, 0.54-0.97; p=0.032) than the low-to-moderate-intensity statin group. However, intriguingly, in elderly patients, the efficacy for reducing adverse clinical outcomes between two statin-intensity groups did not differ (TVF : aHR, 1.1; 95% CI, 0.76-1.59; p=0.63; TLR : aHR, 0.76; 95% CI, 0.37-1.57; p=0.46) (Table 3).

The forest plot of hazard ratio for adverse clinical outcomes from 1 month to 12 months was shown in Figure 4. No significant treatment interactions were detected in subgroups defined by aged of 75 (Pinteraction = 0.10 for TVF, Pinteraction = 0.15 for CV death, and Pinteraction = 0.78 for TLR). Kaplan-Meier curves for the secondary end points were also shown in Appendix Figure 2.

Modified text :

The incidence rates of clinical outcomes between two statin-intensity groups in the overall population, younger and elderly patients in maintenance phase are shown in Table 4. Kaplan-Meier curves for TVF and secondary end points in maintenance phase were shown in Figure 3 and Appendix Figure 2, which were adjusted by IPTW analysis. In overall population, the risk of adverse clinical outcomes according to statin-intensity was not different except for TLR (TVF : aHR, 0.85; 95% CI, 0.69-1.05; p=0.12; TLR : aHR, 0.73; 95% CI, 0.55-0.96; p=0.022; Table 4). In younger patients, the high-intensity statin group showed significantly better clinical outcomes in terms of TVF (aHR, 0.76; 95% CI, 0.59-0.99; p=0.038) and TLR (aHR, 0.72; 95% CI, 0.54-0.97; p=0.032) than the low-to-moderate-intensity statin group. However, intriguingly, in elderly patients, the efficacy for reducing adverse clinical outcomes between two statin-intensity groups did not differ (TVF : aHR, 1.1; 95% CI, 0.76-1.59; p=0.63; TLR : aHR, 0.76; 95% CI, 0.37-1.57; p=0.46). These findings were unchanged after IPTW adjustment for differences in baseline covariates.

The forest plot of hazard ratio after IPTW analysis for adverse clinical outcomes from 1 month to 12 months was shown in Figure 4. No significant treatment interactions were detected in subgroups defined by aged of 75 (Pinteraction = 0.18 for TVF, Pinteraction = 0.35 for all-cause death, Pinteraction = 0.15 for CV death, and Pinteraction = 0.76 for TLR).

Original text : Figure 3

Modified text :

Figure 3. Adjusted Kaplan-Meier curves for the primary end point according to statin-intensity in younger and elderly patients using inverse probability weighting.

 Original text : Results, Line 245.

The forest plot of hazard ratio for adverse clinical outcomes from 1 month to 12 months was shown in Figure 4. No significant treatment interactions were detected in subgroups defined by aged of 75 (Pinteraction = 0.10 for TVF, Pinteraction = 0.15 for CV death, and Pinteraction = 0.78 for TLR). Kaplan-Meier curves for the secondary end points were also shown in Appendix Figure 2.

 Modified text : 

 The forest plot of hazard ratio after IPTW analysis for adverse clinical outcomes from 1 month to 12 months was shown in Figure 4. No significant treatment interactions were detected in subgroups defined by aged of 75 (Pinteraction = 0.18 for TVF, Pinteraction = 0.35 for all-cause death, Pinteraction = 0.15 for CV death, and Pinteraction = 0.76 for TLR).

Original text : Figure legends, Line 550.

 Modified text :

Figure 4.

Effects of statin intensity on clinical outcomes in maintenance phase after IPTW analysis, subdivided by aged of 75.

*derived from unmatched population

**aHR, adjusted hazard ratio for high-intensity statin treatment compared with less intensive strategy after IPTW analysis.

TVF, target-vessel failure; TLR, target lesion revascularization; HR, hazard ratio; NA, not applicable.

Figure 4.

3. The primary outcome needs to be clarified – is it a composite of TVF, CV death, TV-MI, TLR, or are these co-primary? If they are co-primary, there needs to be correction for multiple testing. If it is a composite, the main results should be interpreted as the composite and only secondarily should individual findings be highlighted. Similarly, findings should be tempered accordingly.

Answer : Thank you for your comment. In methods session, we clarified the primary outcome as a composite. As you recommended, we described primary and secondary outcomes separately in results session. To convey a clearer message, unnecessary secondary outcomes were omitted. Also, we changed the previous figure for K-M curves to make it easier for readers to understand.

 Original text : Methods, Line 133

The primary end point of the study was target-vessel failure (TVF), defined as a composite of cardiovascular (CV) death, target-vessel MI (TV-MI), or target lesion revascularization (TLR). The secondary end points were included the individual components of primary end point, any death, any MI, any repeat revascularization (RR), patient-oriented clinical event (POCE), defined as a composite of any death, any MI, or any RR.

 Modified text :

The primary end point of the study was target-vessel failure (TVF), defined as a composite of cardiovascular (CV) death, target-vessel MI (TV-MI), or target lesion revascularization (TLR). The secondary end points included the individual components of primary end point, and any death.

 Original text : Results, Line 224

 Kaplan-Meier curves for TVF and TLR at 1 month were shown in Figure 3.

 Modified text :

Kaplan-Meier curves for TVF at 1 month were shown in Figure 3, adjusted by IPTW analysis.

 Original text : Appendix Figure 2.

 Modified text : 

Appendix Figure 2. Adjusted Kaplan-Meier curves for the secondary end points from 1 month to 12 months using inverse probability weighting.

A-B) Cardiovascular death; C-D) Target vessel myocardial infarction; E-F) Target lesion revascularization.

Left panels (A,C,E) represented clinical outcomes in younger patients (<75 years old) and right panels (B,D,F) in elderly patients (≥75 years old).

CV, cardiovascular; TV-MI, target vessel myocardial infarction; TLR, target lesion revascularization.

Minor: Avoid the use of trial language – this is an observational study and associations are being studied, not effects or efficacy.

Answer : Thank you for your advice. We corrected the expression ‘effects or efficacy’ throughout the manuscript.

 Original text : Abstract, Line 51 and 63.

 Modified text :

Methods Using the COREA-AMI registry, we sought to compare the clinical impact of high- versus low-to-moderate-intensity statin in younger (<75 years old) and elderly (≥75 years old) patients. Of 10,719 patients, we included 8,096 patients treated with drug-eluting stents. All patients were classified into high-intensity versus low-to-moderate-intensity statin group according to statin type and dose at discharge. The primary end point was target-vessel failure (TVF), a composite of cardiovascular death, target-vessel MI, or target-lesion revascularization (TLR) from 1 month to 12 months after index PCI.

Conclusions In this AMI cohort underwent PCI, high-intensity statin showed the better 1-year clinical outcomes than low-to-moderate-intensity statin in younger patients. Meanwhile, the incidence rates of adverse clinical events between high- and low-to-moderate-intensity statin were not statistically different in elderly patients. Further randomized study with large elderly population is warranted.

 Original text : Results, Line 241. 

 Modified text :

However, intriguingly, in elderly patients, the incidence rates of adverse clinical outcomes between two statin-intensity groups did not differ (TVF : aHR, 1.1; 95% CI, 0.76-1.59; p=0.63; TLR : aHR, 0.76; 95% CI, 0.37-1.57; p=0.46).

Original text : Results, Line 277.

Modifed text : 

However, in elderly patients, the incidence rates of adverse clinical outcomes did not differ between continuous use of high- and low-to-moderate-intensity statin (aHR 0.7, 95% CI 0.28-1.77, p=0.46).

Original text : Discussion, Line 298.

Modified text :

4) in contrast, the incidence rates of adverse clinical events between high- and low-to-moderate-intensity statin were not statistically different in elderly patients.

Original text : Conclusion, Line 396

Modified text :

In this AMI cohort underwent PCI, high-intensity statin showed better clinical outcomes at 1-year than low-to-moderate intensity statin in younger patients. Meanwhile, in elderly patients, Meanwhile, the incidence rates of adverse clinical events were not statistically different between high- and low-to-moderate-intensity statin in elderly patients. Further randomized study with large elderly population is warranted.

---

## [Decision Letter · Decision Letter 3]

3 Mar 2022

PONE-D-21-14585R3Clinical Impact of Statin Intensity According to Age in Patients with Acute Myocardial InfarctionPLOS ONE

Dear Dr. Park,

Thank you for submitting your manuscript to PLOS ONE. After careful consideration, we feel that it has merit but does not fully meet PLOS ONE’s publication criteria as it currently stands. Therefore, we invite you to submit a revised version of the manuscript that addresses the points raised during the review process.

Please review the comments for minor adjustments from the reviewers above, and address each one individually prior to resubmission. We thank you for your time and attention to these items. 

We look forward to receiving your revised manuscript.

Kind regards,

R. Jay Widmer

Academic Editor

PLOS ONE

Journal Requirements:

Reviewers' comments:

Reviewer's Responses to Questions

**Comments to the Author**

1. If the authors have adequately addressed your comments raised in a previous round of review and you feel that this manuscript is now acceptable for publication, you may indicate that here to bypass the “Comments to the Author” section, enter your conflict of interest statement in the “Confidential to Editor” section, and submit your "Accept" recommendation.

Reviewer #3: All comments have been addressed

Reviewer #4: All comments have been addressed

Reviewer #5: (No Response)

2. Is the manuscript technically sound, and do the data support the conclusions?

Reviewer #3: Yes

Reviewer #4: Yes

Reviewer #5: Yes

3. Has the statistical analysis been performed appropriately and rigorously? 

Reviewer #3: Yes

Reviewer #4: Yes

Reviewer #5: Yes

4. Have the authors made all data underlying the findings in their manuscript fully available?

Reviewer #3: Yes

Reviewer #4: Yes

Reviewer #5: Yes

5. Is the manuscript presented in an intelligible fashion and written in standard English?

Reviewer #3: Yes

Reviewer #4: Yes

Reviewer #5: Yes

6. Review Comments to the Author

Reviewer #3: All my concerns have been adequately addressed.

This is a nice contribution to the literature.

Reviewer #4: The author analyzed clinical impact of statin intensity as secondary prevention in patients with acute myocardial infarction (AMI) using large, prospective, multicenter registry data. Patients were divided who treated percutaneous coronary intervention (PCI) due to AMI into taking high-intensity statin or low to moderate-intensity statin based on the age at which the 75-years old. The propensity score matching and inverse probability of treatment weighting methods were applied to properly compare the two groups, and as a consequence, the two groups could be compared relatively evenly. 1-year clinical outcomes showed that groups under 75 years old who took a high-intensity statin had better result than low-to-moderate-intensity statin group. However, in the groups which over 75-years old and overall groups show no significant difference according to intensity of statin doses. Although the characteristics of the patients subject to this study were heterogenous, comparison were made possible through appropriate statistical methods. The conclusion and discussion of this manuscript can help clinical direction in statin use as secondary preventions. However, there are several issues that should be adequately revised.

1. If patients have an LDL value during a 1-year follow-up periods, it will be feasible to determine how much the LDL level fluctuated for each statin group and explain the association by analyzing for a lowering trend in multiple event rates and LDLs.

2. It would be more helpful to understand the background of the hypothesis to have an explanation or evidence for the criteria for dividing the age group by 75-years old

Reviewer #5: General Comments:

The manuscript entitled "Clinical impact of statin intensity according to age in patients with acute myocardial infarction" was reviewed. This study is a substudy from a large prospective multicenter AMI registry study in South Korea, and it shows the lack of benefit of strong statins in the elderly AMI patients. This is a very interesting study using a large volume database to compare the clinical impact of high- versus low-to-moderate-intensity statin in younger and elderly patients, has not been properly proven in AMI registry so far. In general, this manuscript is very well written. However, the reviewer would like to comment on the following two points.

Specific Comments:

1. This study used a 10-year database from 2004 to 2014, but even in the span of 10 years, there has been a further aging of the population and changes in dietary habits. In addition, the guidelines for statin administration have changed from “fire and forget” to “target to treat”, and the treatment policies of interventional cardiologists are also likely to have changed. Are there any changes in the results between the first 5 years and the second 5 years in this study?

2. Based on the results of this study, is it more likely that cholesterol absorption inhibition, such as ezetimibe, will be effective in elderly patients rather than statin strength?

7. PLOS authors have the option to publish the peer review history of their article (what does this mean?). If published, this will include your full peer review and any attached files.

Reviewer #3: No

Reviewer #4: **Yes: **Duk-Woo Park

Reviewer #5: No

---

## [Author Response · Author response to Decision Letter 3]

16 Apr 2022

Reviewer #3: All my concerns have been adequately addressed.

This is a nice contribution to the literature.

 ANSWER : Thank you for your generous understanding. Thanks.

Reviewer #4: The author analyzed clinical impact of statin intensity as secondary prevention in patients with acute myocardial infarction (AMI) using large, prospective, multicenter registry data. Patients were divided who treated percutaneous coronary intervention (PCI) due to AMI into taking high-intensity statin or low to moderate-intensity statin based on the age at which the 75-years old. The propensity score matching and inverse probability of treatment weighting methods were applied to properly compare the two groups, and as a consequence, the two groups could be compared relatively evenly. 1-year clinical outcomes showed that groups under 75 years old who took a high-intensity statin had better result than low-to-moderate-intensity statin group. However, in the groups which over 75-years old and overall groups show no significant difference according to intensity of statin doses. Although the characteristics of the patients subject to this study were heterogenous, comparison were made possible through appropriate statistical methods. The conclusion and discussion of this manuscript can help clinical direction in statin use as secondary preventions. However, there are several issues that should be adequately revised.

1. If patients have an LDL value during a 1-year follow-up periods, it will be feasible to determine how much the LDL level fluctuated for each statin group and explain the association by analyzing for a lowering trend in multiple event rates and LDLs.

 ANSWER : Thank you for your valuable comment. We reviewed the raw data for LDL value during follow-up period. However, we found that 3,585 (44%) of follow-up LDL value were existed and the rest were missing value. Since most of follow-up LDL value were measured only once, we could not assess an LDL level fluctuation but could check a trend. In younger patients, high-intensity statin group showed more proportion of LDL level less than 70 mg/dL. In contrast, in elderly patients, there was no statistical difference according to statin-intensity. We could show that in the Table below. If the reviewer wants, it will be reflected in the manuscript.

Table.

2. It would be more helpful to understand the background of the hypothesis to have an explanation or evidence for the criteria for dividing the age group by 75-years old.

 ANSWER : I appreciate your comment. Statin therapy has been shown to reduce major vascular events and vascular mortality in patients with established ASCVD. However, rates of use of statin have been shown to decline with increasing age, and are substantially lower in elderly patients (> 75 years old), reflecting differences in both prescribing and compliance. One explanation might be that, there is a lack of data regarding the efficacy and safety of high-intensity statin in the elderly patients (> 75 years old), mainly because this patient group is often underrepresented in RCT comparing statin intensity. Also, current guidelines divided the age group by 75-years old. Therefore, we followed the definition of elderly patients as current guidelines did.

 Original text : Background, Line 92

 However, there is a lack of data regarding the efficacy and safety of high-intensity statin in the elderly patients (≥75 years old), mainly because this patient group is often underrepresented in randomised controlled trials comparing statin intensity.

 Modified text : 

 However, prescription rates of statin have been shown to be declined with increasing age, and are lower, especially in elderly patients (>75 years old).[14] One explanation might be that, there has been a lack of data regarding the efficacy and safety of high-intensity statin in the elderly patients (≥75 years old), mainly because this patient group is often underrepresented in randomised controlled trials comparing statin intensity. For this reason, the level of evidence of using a high-intensity statin for elderly patients is lower compared to that for younger patients in current guidelines,[11, 12] of which elderly patients were divided the age goup by 75-years old. Moreover, there are no studies comparing the efficacy of statin intensity according to age in AMI patients treated by PCI using drug-eluting stents (DESs).

Reference

14. Koopman C, Vaartjes I, Heintjes EM, Spiering W, van Dis I, Herings RM, et al. Persisting gender differences and attenuating age differences in cardiovascular drug use for prevention and treatment of coronary heart disease, 1998-2010. European heart journal. 2013;34(41):3198-205. Epub 2013/09/21. doi: 10.1093/eurheartj/eht368. PubMed PMID: 24046432.

Reviewer #5: General Comments:

The manuscript entitled "Clinical impact of statin intensity according to age in patients with acute myocardial infarction" was reviewed. This study is a substudy from a large prospective multicenter AMI registry study in South Korea, and it shows the lack of benefit of strong statins in the elderly AMI patients. This is a very interesting study using a large volume database to compare the clinical impact of high- versus low-to-moderate-intensity statin in younger and elderly patients, has not been properly proven in AMI registry so far. In general, this manuscript is very well written. However, the reviewer would like to comment on the following two points.

Specific Comments:

1. This study used a 10-year database from 2004 to 2014, but even in the span of 10 years, there has been a further aging of the population and changes in dietary habits. In addition, the guidelines for statin administration have changed from “fire and forget” to “target to treat”, and the treatment policies of interventional cardiologists are also likely to have changed. Are there any changes in the results between the first 5 years and the second 5 years in this study?

 ANSWER : Thank you for your comment. We additionally analyzed the result divided by 5 years (2004-2008 and 2009-2014). Interestingly, in first 5 years, there is few patients who were prescribed high-intensity statin both in younger and elderly patients. Even there was no statistical significance due to underpower, the trend was similar with our original results. However, this result would mislead our readers. If the reviewer wants, it will be reflected in the manuscript.

Table. Event rates and hazard ratios for clinical outcomes divided by 5 years.

2. Based on the results of this study, is it more likely that cholesterol absorption inhibition, such as ezetimibe, will be effective in elderly patients rather than statin strength?

 ANSWER : I appreciate your valuable comment. As we discussed this issue in the discussion session, the addition of ‘ezetimibe’, which inhibits cholesterol absorption, would be effective in elderly patients (from the results of DEBATE, IMPROVE-IT study). However, our study was designed to compare the statin-intensity. Thus, we cannot concede or refute your comment based on our study design and result.

---

## [Decision Letter · Decision Letter 4]

19 May 2022

Clinical Impact of Statin Intensity According to Age in Patients with Acute Myocardial Infarction

PONE-D-21-14585R4

Dear Dr. Park,

We’re pleased to inform you that your manuscript has been judged scientifically suitable for publication and will be formally accepted for publication once it meets all outstanding technical requirements.

Kind regards,

R. Jay Widmer

Academic Editor

PLOS ONE

Additional Editor Comments (optional):

Reviewers' comments:

Reviewer's Responses to Questions

**Comments to the Author**

1. If the authors have adequately addressed your comments raised in a previous round of review and you feel that this manuscript is now acceptable for publication, you may indicate that here to bypass the “Comments to the Author” section, enter your conflict of interest statement in the “Confidential to Editor” section, and submit your "Accept" recommendation.

Reviewer #3: All comments have been addressed

Reviewer #4: All comments have been addressed

Reviewer #5: All comments have been addressed

Reviewer #6: (No Response)

2. Is the manuscript technically sound, and do the data support the conclusions?

Reviewer #3: Yes

Reviewer #4: Yes

Reviewer #5: Yes

Reviewer #6: Yes

3. Has the statistical analysis been performed appropriately and rigorously? 

Reviewer #3: Yes

Reviewer #4: Yes

Reviewer #5: Yes

Reviewer #6: Yes

4. Have the authors made all data underlying the findings in their manuscript fully available?

Reviewer #3: Yes

Reviewer #4: Yes

Reviewer #5: Yes

Reviewer #6: Yes

5. Is the manuscript presented in an intelligible fashion and written in standard English?

Reviewer #3: Yes

Reviewer #4: Yes

Reviewer #5: Yes

Reviewer #6: No

6. Review Comments to the Author

Reviewer #3: If the authors can include the LDL analysis as an appendix, recognizing the limitation of missing data, I believe this would be of interest to readers.

Reviewer #4: Revised manuscript has been substantially improved.

All issues have been well addressed. We have no further comments.

Reviewer #5: (No Response)

Reviewer #6: Dear authors,

I have read the manuscript and it is overall scientifically sound.

In the interest of both the authors and the journal, I notice that PLOS ONE does not copyedit accepted manuscripts and it may therefore benefit to request a version of the manuscript that has been carefully edited since there remains several "asienglish" grammatical errors, which are annoying for the readers.

7. PLOS authors have the option to publish the peer review history of their article (what does this mean?). If published, this will include your full peer review and any attached files.

Reviewer #3: No

Reviewer #4: No

Reviewer #5: No

Reviewer #6: No

---

## [Editor Report · Acceptance letter]

3 Jun 2022

PONE-D-21-14585R4 

Clinical impact of statin intensity according to age in patients with
acute myocardial infarction 

Dear Dr. Park:

I'm pleased to inform you that your manuscript has been deemed suitable for publication in PLOS ONE. Congratulations! Your manuscript is now with our production department. 

Kind regards, 

on behalf of

Dr. R. Jay Widmer 

Academic Editor

PLOS ONE